# Continuous Temporal Domain Generalization

**Zekun Cai**[1,4], **Guangji Bai**[2], **Renhe Jiang**[1*], **Xuan Song**[3,4], and **Liang Zhao**[2]

[1]The University of Tokyo, Tokyo, Japan
[2]Emory University, Atlanta, GA, USA
[3]Jilin University, Changchun, China
[4]Southern University of Science and Technology, Shenzhen, China
`{caizekun,jiangrh,songxuan}@csis.u-tokyo.ac.jp`
`{guangji.bai,liang.zhao}@emory.edu`

## Abstract

Temporal Domain Generalization (TDG) addresses the challenge of training predictive models under temporally varying data distributions. Traditional TDG approaches typically focus on domain data collected at fixed, discrete time intervals, which limits their capability to capture the inherent dynamics within continuous-evolving and irregularly-observed temporal domains. To overcome this, this work formalizes the concept of Continuous Temporal Domain Generalization (CTDG), where domain data are derived from continuous times and are collected at arbitrary times. CTDG tackles critical challenges including: 1) Characterizing the continuous dynamics of both data and models, 2) Learning complex high-dimensional nonlinear dynamics, and 3) Optimizing and controlling the generalization across continuous temporal domains. To address them, we propose a Koopman operator-driven continuous temporal domain generalization (Koodos) framework. We formulate the problem within a continuous dynamic system and leverage the Koopman theory to learn the underlying dynamics; the framework is further enhanced with a comprehensive optimization strategy equipped with analysis and control driven by prior knowledge of the dynamics patterns. Extensive experiments demonstrate the effectiveness and efficiency of our approach. The code can be found at: `https://github.com/Zekun-Cai/Koodos`.

## 1 Introduction

In practice, the distribution of training data often differs from that of test data, leading to a failure in generalizing models outside their training environments. Domain Generalization (DG) [49; 46; 21; 14; 40] is a machine learning strategy designed to learn a generalized model that performs well on the unseen target domain. The task becomes particularly pronounced in dynamic environments where the statistical properties of the target domains change over time [16; 33; 4], prompting the development of Temporal Domain Generalization (TDG) [27; 39; 41; 2; 55; 57]. TDG recognizes that domain shifts are temporally correlated. It extends DG approaches by modeling domains as a sequence rather than as categorical entities, making it especially beneficial in fields where data is inherently time-varying.

Existing works in TDG typically concentrate on the discrete temporal domain, where domains are defined by distinct "points in time" with fixed time intervals, such as second-by-second data (Rot 2-Moons [48]) and annual data (Yearbook [54]). In this framework, data bounded in a time interval are considered as a detached domain, and TDG approaches primarily employ probabilistic models to predict domain evolutions. For example, LSSAE [41] employs a probabilistic generative model to analyze latent structures within domains; DRAIN [2] builds a Bayesian framework to predict future model parameters, and TKNets [57] constructs domain transition matrix derived from the data.

---

* Corresponding author

38th Conference on Neural Information Processing Systems (NeurIPS 2024).

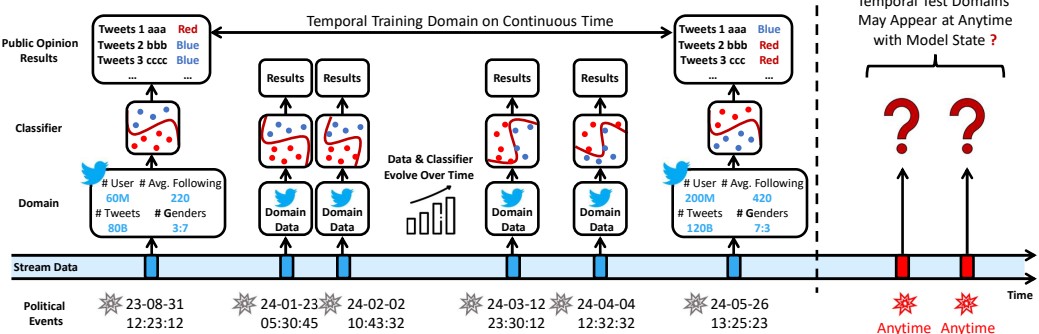

Figure 1: An example of continuous temporal domain generalization. Consider training classification models for public opinion prediction via tweets, where the training domains are only available at specific political events (e.g., presidential debates), we wish to generalize the model to any future based on the underlying data distribution drift within the time-irregularly distributed training domains.

However, in practice, data may not always occur or be observed at discrete, regularly spaced time points. Instead, events and observations unfold irregularly and unpredictably in the time dimension, leading to temporal domains distributed irregularly and sparsely over continuous time. Formally, this paper introduces a problem termed Continuous Temporal Domain Generalization (CTDG), where both seen and unseen tasks reside in different domains at continuous, irregular time points. Fig. 1 illustrates an example of public opinion prediction after political events via Twitter data. Unlike the assumption in traditional TDG that the temporal regularly domains, the data are collected for the times near political events that may occur in arbitrary times. In the meanwhile, the domain data evolve constantly and continuously over time, e.g., active users increase, new friendships are formed, and age and gender distribution changes. Correspondingly, the ideal classifier should gradually change with the domain at random moments to counter the data distribution change over time. Finally, we are concerned with the state of the predictive model at any moment in the future. CTDG is ubiquitous in other fields. For example, in disaster management, relevant data are collected during and following disasters, which may occur at any time throughout the year. In healthcare, critical information about diagnosis and treatment is typically only documented during episodes of care rather than evenly throughout the lifetime. Hence, the CTDG task necessitates the characterization of the continuous dynamics of irregular time-distributed domains, which cannot be handled by existing TDG methods designed for discrete-dynamics and fixed-interval times.

Despite the importance of CTDG, it is still a highly open research area that is not well explored because of several critical hurdles in: **1) Characterizing data dynamics and their impact on model dynamics.** The irregular times of the temporal domains require us to characterize the continuous dynamics of the data and, hence, the model dynamics ultimately. However, the continuous-time data dynamics are unknown and need to be learned across arbitrary time points. Furthermore, it is imperative yet challenging to know how the model evolves according to the data dynamics in continuous times. Therefore, we don't have a direct observation of the data dynamics and the model dynamics we want to learn, which prevents us from existing continuous time modeling techniques. **2) Learning the underlying dynamics of over-parametrized models.** Deep neural networks (e.g., Multi-Layer Perceptron and Convolutional Neural Network) are highly nonlinear and over-parametrized, and hence, the evolutionary dynamics of model states over continuous time are high-dimensional and nonlinear. Consequently, the principal dynamics reside in a great number of latent dimensions. Properly representing and mapping these dynamics into a learnable space remains a challenge. **3) Jointly optimizing the model and its dynamics under possible inductive bias.** The model learning for individual domains will be entangled with the learning of the continuous dynamics across these models. Furthermore, in many situations, we may have some high-level prior knowledge about the dynamics, such as whether there are convergent, divergent, or periodic patterns. It is an important yet open topic to embed them into the CTDG problem-solving.

To address all the challenges, we propose the Koopman operator-driven continuous temporal domain generalization framework (Koodos). Specifically, the Koodos framework articulates the evolutionary continuity of the predictive models in CTDG, then leverages a continuous dynamic approach to model its smooth evolution over time. Koodos further simplifies the nonlinear model system by projecting them into a linearized space via the Koopman Theory. Finally, Koodos provides an interface that

reveals the internal model dynamic characteristics, as well as incorporates prior knowledge and constraints directly into the joint learning process.

## 2 Related works

**Domain Generalization (DG) and Domain Adaptation (DA).** DG approaches attempt to learn a model from multiple source domains that generalize well to an unseen domain [38; 37; 28; 5; 13; 56]. Existing DG methods can be classified into three strategies [49]: (1) Data manipulation techniques, such as data augmentation [45; 46; 57] and data generation [32; 40]; (2) Representation learning focuses on extracting domain-invariant features [17; 18] and disentangling domain-shared from domain-specific features [30]; (3) Learning strategies encompass ensemble learning [34], meta-learning [26; 14; 9], and gradient-based approaches [21]. Unlike DG, DA methods require simultaneously accessing source and target domain data to facilitate alignment and adaptation [50; 51; 29]. The technique includes domain-invariant learning [17; 47; 35; 48; 53], domain mapping [6; 20; 15; 31], ensemble methods [43], and so on. *Both DG and DA are limited to considering generalization across categorical domains, which treats domains as individuals but ignores the smooth evolution of them over time.*

**Temporal Domain Generalization (TDG).** TDG is an emerging field that extends traditional DG techniques to address challenges associated with time-varying data distributions. TDG decouples time from the domain and constructs domain sequences to capture its evolutionary relationships. S-MLDG [27] pioneers a sequential domain DG framework based on meta-learning. Gradient Interpolation (GI) [39] proposes to extrapolate the generalized model by supervising the first-order Taylor expansion of the learned function. LSSAE [41] deploys a probabilistic framework to explore the underlying structure in the latent space of predictive models. DRAIN [2] constructs a recurrent neural network that dynamically generates model parameters to adapt to changing domains. TKNets [57] minimize the divergence between forecasted and actual domain data distributions to capture temporal patterns.

Despite these studies, traditional TDG methods are limited to requiring domains presented in discrete time, which disrupts the inherent continuity of changes in data distribution, and the generalization can only be carried forward by very limited steps. *No work treats time as a continuous variable, thereby failing to capture the full dynamics of evolving domains and generalize to any moment in the future.*

**Continuous Dynamical Systems (CDS).** CDS are fundamental in understanding how systems evolve without the constraints of discrete intervals. They are the study of the dynamics for systems defined by differential equations. The linear multistep method or the Runge-Kutta method can solve the Order Differential Equations (ODEs) [19]. Distinguishing from traditional methods, Neural ODEs [11] represent a significant advancement in the field of CDS. It defines a hidden state as a solution to the ODEs initial-value problem, and parameterizes the derivatives of the hidden state using a neural network. The hidden state can then be evaluated at any desired time using a numerical ODEs solver. Many recent studies have proposed on which to learn differential equations from data [42; 22; 23; 36].

## 3 Problem definition

**Continuous Temporal Domain Generalization (CTDG):** We address prediction tasks where the data distribution evolves over time. In predictive modeling, a domain $\mathcal{D}(t)$ is defined as a dataset collected at time $t$ consisting of instances $\{(x_i^{(t)}, y_i^{(t)}) \in \mathcal{X}(t) \times \mathcal{Y}(t)\}_{i=1}^{N(t)}$, where $x_i^{(t)}$, $y_i^{(t)}$ and $N(t)$ represent the feature, target and the number of instances at time $t$, and $\mathcal{X}(t)$, $\mathcal{Y}(t)$ denote the input feature space and label space at time $t$, respectively. We focus on the existence of gradual concept drift across continuous time, indicating that domain conditional probability distributions $P(Y(t)|X(t))$, with $X(t)$ and $Y(t)$ representing the random variables for features and targets at time $t$, change smoothly and seamlessly over continuous time like streams without abrupt jumps.

During training, we are provided with a sequence of observed domains $\{\mathcal{D}(t_1), \mathcal{D}(t_2), \ldots, \mathcal{D}(t_T)\}$ collected at arbitrary times $\mathcal{T} = \{t_1, t_2, \ldots, t_T\}$, where $t_i \in \mathbb{R}^+$ and $t_1 < t_2 < \ldots < t_T$. For each domain $\mathcal{D}(t_i)$ at time $t_i \in \mathcal{T}$, we learn the predictive model $g(\cdot; \theta(t_i)) : \mathcal{X}(t_i) \to \mathcal{Y}(t_i)$, where $\theta(t_i)$ denotes the parameters of function $g$ at timestamp $t_i$. We model the dynamics across the parameters $\{\theta(t_1), \theta(t_2), \ldots, \theta(t_T)\}$, and finally predict the parameters $\theta(s)$ for the predictive model $g(\cdot; \theta(s)) : \mathcal{X}(s) \to \mathcal{Y}(s)$ at any given time $s \notin \mathcal{T}$. For simplicity, in subsequent, we will use $\mathcal{D}_i$, $X_i, Y_i, \theta_i$ to represent $\mathcal{D}(t_i), X(t_i), Y(t_i), \theta(t_i)$ at time $t_i$.

Unlike traditional temporal domain generalization approaches [39; 2; 52; 57] that divide time into discrete intervals and require domain data be collected at fixed time steps, the CTDG problem

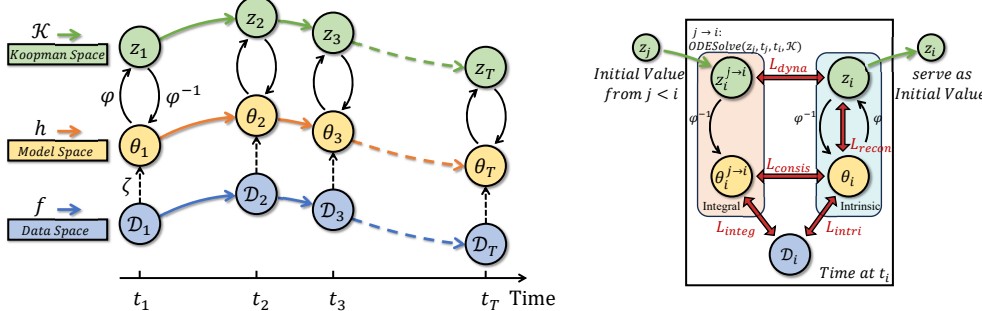

(a) Three Spaces and Parallel Flows in Continuous Temporal Domains    (b) State Constraints at a Given Moment

Figure 2: Macro-flows and micro-constraints in the proposed model framework.

treats time as continuous variable, allowing for the time points in training and test set to be any arbitrary positive real numbers, and requiring models to deal with the temporal domains as continuous streams. CTDG problem poses several unprecedented, substantial challenges in: 1) Characterizing the continuous dynamics of data and how that decides model dynamics; 2) Modeling the low-dimensional continuous dynamics of the predictive model, which are embedded in high-dimensional space due to over-parametrization; and 3) Optimizing and analyzing the continuous predictive model system, including the application of inductive biases to control its behavior.

## 4 Methodology

In this section, we outline our strategies to tackle the problem of CTDG by overcoming its substantial challenges. First, to learn the continuous drifts of data and models, we synchronize between the evolving domain and the model by establishing the continuity of the model parameters over time, which is ensured through the formulation of differential equations as elaborated in Section 4.1. To fill the gap between high-dimensional model parameters and low-dimensional model continuous dynamics, we simplify the representation of complex dynamics into a principal, well-characterized Koopman space as is detailed in Section 4.2. To jointly learn the model and its dynamics under additional inductive bias, in Section 4.3, we design a series of organized loss functions to form an efficient end-to-end optimization strategy. Overall, as shown in Fig. 2(a), there are three dynamic flows in our system, which are the Data Flow, the Model Flow, and the Koopman Representation Flow. Through the proposed framework, we aim to ensure not only that the model responds to the statistical variations inherent in the data, but also that the characteristics of the three flows are consistent.

### 4.1 Characterizing the continuous dynamics of the data and the model

In this section, we explore the relationship between the evolving continuous domains and the corresponding dynamics of the predictive models. We demonstrate that the dynamics of temporal domains lead to the internal update of the predictive model continuously over time in Theorem 1. Following this, we develop a learnable dynamic system for the continuous state of the model by synchronizing the behaviors of the domain and model.

**Assumption 1.** *Consider the gradual concept drift within the continuous temporal domains. It is assumed that the conditional probability distribution $P_t(Y|X)$ changes continuously over time, and its dynamics are characterized by a function $f$, which models the variations in the distribution.*

**Theorem 1.** *(Continuous Evolution of Model Parameters) Given Assumption 1, it follows that the parameters $\theta_t$ of the predictive model $g(\cdot; \theta_t)$ also evolve continuously over time, and its dynamics are jointly determined by the current state of the model and the function $f$.*

*Proof.* The temporal derivative of the functional space $g(\cdot; \theta_t)$ represents its evolution in direct response to the changes in the conditional probability distribution. Without loss of generality, the ground-truth functional space can be modeled by an ordinary differential equation:

$$dg(\cdot; \theta_t)/dt = f(g(\cdot; \theta_t), t). \tag{1}$$

Applying the chain rule to decompose the temporal derivative of $g$:

$$\frac{d}{dt}g(\cdot; \theta_t) = \sum_{i=1}^{n} \frac{\partial g}{\partial \theta_{t,i}} \frac{d\theta_{t,i}}{dt} = J_g(\theta_t) \frac{d\theta_t}{dt}, \tag{2}$$

where $J_g(\theta_t)$ is the Jacobian matrix of $g$ with respect to $\theta_t$.

By equating the Eq. 2 to the expression involving $f$:

$$J_g(\theta_t)\frac{d\theta_t}{dt} = f(g(\cdot;\theta_t),t). \tag{3}$$

Assuming that $J_g(\theta_t)$ is invertible, the ground-truth derivative of $\theta_t$ with respect to time is given by:

$$\frac{d\theta_t}{dt} = J_g(\theta_t)^{-1}f(g(\cdot;\theta_t),t). \tag{4}$$

$\square$

It is known from the Proof that the evolution continuity of $\theta_t$ follows from a differential equation. Eq. 4 identifies a dynamic framework for the predictive model parameters, which provides a strong motivation to develop dynamic systems for them.

The principal challenge arises from the unknown dynamics enclosed in $f$, which prevents getting $\theta_t$ by direct mathematical computation. Recognizing this, we propose a learning-based approach to model the parameter dynamics from the observed domains. We construct a learnable parameter dynamics function $h(\theta_t, t; \phi)$, parameterized by $\phi$, optimized to topological conjugation between the model dynamics and the data dynamics. Topological conjugation guarantees that the parameter's orbits faithfully characterize the underlying data dynamics, thereby ensuring the model's generalization capability across all time. Specifically, as illustrated in Fig. 2(a) of the Data Flow and the Model Flow, assuming a conceptual perfect mapping $\xi$ from the data space to the parameter space, topological conjugation suggests that the composition of the learned dynamic $h$ after $\xi$ should behave identically to applying $\xi$ after the dynamics $f$, i.e., $h \circ \xi = \xi \circ f$ holds, where $\circ$ represents function composition. In the CTDG problem, the objective of conjugation is to optimize the model dynamics $h$ and the predictive parameters $\theta$ simultaneously, to minimize the discrepancy between the dynamically derived parameters from $h$ and those obtained through direct training, formulated as follows:

$$\phi = \arg\min_{\phi} \sum\nolimits_{i=1}^{T} \sum\nolimits_{j=1}^{i} \|\theta_i, \theta_i^{j \to i}\|_2, \tag{5}$$

where $\|\cdot\|_2$ denotes the Euclidean norm, and each $\theta_i$ is determined by:

$$\theta_i = \arg\min_{\theta_i} \mathcal{L}(Y_i, g(X_i; \theta_i)), \tag{6}$$

and $\theta_i^{j \to i}$ is defined as the integral parameters at $t_i$ obtained from $t_j < t_i$:

$$\theta_i^{j \to i} = \theta_j + \int_{t_j}^{t_i} h(\theta_\tau, \tau; \phi)\, d\tau. \tag{7}$$

Here, $\mathcal{L}$ symbolizes the loss function tailored to the prediction task. By employing Eq. 7, the model dynamics are defined for all observation time, and $\phi$ can be obtained by optimizing Eq. 5 and Eq. 6 via gradient descent. After that, the parameters of the predictive model can be calculated at any specific time using ODE solvers.

## 4.2 Modeling nonlinear model dynamics by Koopman operators

The aforementioned learning approaches are, however, limited in efficient modeling and prediction due to the entangled nonlinear dynamics in the high-dimensional parameters space. Identifying strongly nonlinear dynamics through such space is particularly susceptible to overfitting [8], leading to future predictive models eventually diverging from bifurcation to chaos. Despite the complexity of $h$, it is governed by the data dynamics $f$, which are typically low-dimensional and exhibit simple, predictable patterns. This suggests that the governing equations for $h$ are sparse within the over-parameterized space, and thus allow for a more simplified and manageable representation in a properly transformed space. Motivated by this, we propose leveraging the well-established Koopman Theory [24; 7] to represent these complex parameter dynamics. Koopman Theory provides a method for the global linearization of any nonlinear dynamics. It expresses the complex dynamic system as an infinite-dimensional Koopman operator acting on the Hilbert space of the system state measurement functions, in which the nonlinear dynamics will become linearized.

To facilitate this approach, the target is to identify a set of intrinsic coordinates $\varphi$ that spans a Koopman-invariant subspace. As illustrated in Fig. 2(a) of the Model Flow and Koopman Flow, this transformation maps the parameters $\theta$ into a latent space $z = \varphi(\theta)$, with $z$ resides in a low $n$-dimensional space, $\mathcal{Z} = \mathbb{R}^n$, where the dynamics become more linearized. We also aim to find a finite-dimensional approximation of the Koopman operator, denoted as $\mathcal{K} \in \mathbb{R}^{n \times n}$, that acts on the latent space and advances the observation of the state to the future $\mathcal{K} : \mathcal{Z} \to \mathcal{Z}$. After that we have:

$$d\varphi(\theta)/dt = \mathcal{K}\varphi(\theta). \tag{8}$$

The $z$ dynamics are easier to track because they principleize the dynamical system while maintaining its characteristics. It gives us a tighter representation of the parameters within a linear space, which allows us to learn the simple $\mathcal{K}$ operator instead of the complex coupled dynamics $h(\cdot; \phi)$. Following the transformation, the dynamics of the parameters can be expressed by:

$$z_i^{j \to i} = z_j + \int_{t_j}^{t_i} \mathcal{K} z_\tau \, d\tau, \quad \text{where } z_j = \varphi(\theta_j). \tag{9}$$

Finally, an inverse transformation provided by $\theta_i^{j \to i} = \varphi^{-1}(z_i^{j \to i})$ that maps Koopman space back to the original parameter space. The relational among $\theta, \mathcal{K}, \varphi, \varphi^{-1}$ and the Koopman invariant subspace are bounded by a series of loss functions detailed in the next Section.

### 4.3 Joint optimization of model and its dynamics with prior knowledge

We introduce a comprehensive optimization approach designed to ensure that the system accurately captures the dynamics of the data. This process requires the joint optimization of several interconnected components under the optional constraint of additional prior knowledge about the underlying dynamics: the predictive model parameters $\theta_{1:T}$, the transformation functions $\varphi$ and $\varphi^{-1}$, and the Koopman operator $\mathcal{K}$. Our primary objectives are threefold: *1) Ensuring high prediction accuracy, 2) Maintaining consistency of parameters across different representations and transformations, and 3) Learning the Koopman invariant subspaces effectively*. Fig. 2(b) illustrates the role of each constraint within our system: we manage two sets of states, intrinsic and integral, aligning across three spaces.

**Predictive Model Parameters** ($\theta$): Each observation time corresponds to a predictive model, which is tasked with making predictions and serving as the initial values for the dynamical system. They are primarily optimized to minimize the prediction error $L_{intri}$ of different domains:

$$L_{intri} = \sum_{i=1}^{T} \mathcal{L}(Y_i, g(X_i; \theta_i)). \tag{10}$$

**Koopman Network Parameters** ($\varphi, \varphi^{-1}, \mathcal{K}$): We estimate a function that transforms the parameters of the predictive model into Koopman space. Meanwhile, these predictive model parameters must be stable and consistent when converted between representations. This is realized by three constraints:

1. **Reconstruction Loss:** An autoencoder is leveraged to map parameter space to Koopman space by encoder $\varphi$ and back to the original space by decoder $\varphi^{-1}$. $L_{recon}$ ensures consistency between $\theta$ and its reconstructed form via transformations:

$$L_{recon} = \sum_{i=1}^{T} \|\theta_i, \varphi^{-1}(\varphi(\theta_i))\|_2. \tag{11}$$

2. **Dynamic Fidelity Loss:** This term ensures that the transformation produces a Koopman invariant subspaces in which the dynamics can forward correctly. It measures the fit of the Koopman operator $\mathcal{K}$ against the transferred parameter:

$$L_{dyna} = \sum_{i=1}^{T} \sum_{j=1}^{i} \|z_i, z_i^{j \to i}\|_2, \tag{12}$$

where $z_i = \varphi(\theta_i)$ and $z_i^{j \to i} = z_j + \int_{t_j}^{t_i} \mathcal{K} z_\tau \, d\tau$.

3. **Consistency Loss:** It measures the consistency between the original and the dynamic parameter in the model space:

$$L_{consis} = \sum_{i=1}^{T} \sum_{j=1}^{i} \|\theta_i, \theta_i^{j \to i}\|_2, \quad \text{where } \theta_i^{j \to i} = \varphi^{-1}(z_i^{j \to i}) \tag{13}$$

Additionally, we load the dynamically integral parameters $\theta_i^{j \to i}$ back into the predictive model to evaluate its predictive capability, quantified by $L_{integ} = \sum_{i=1}^{T} \sum_{j=1}^{i} \mathcal{L}(Y_i, g(X_i; \theta_i^{j \to i}))$. Finally, the system optimizes the following combined loss to refine all components simultaneously:

$$\{\theta_{1:T}, \varphi, \varphi^{-1}, \mathcal{K}\} = \underset{\theta_{1:T}, \varphi, \varphi^{-1}, \mathcal{K}}{\arg min} \; (\alpha L_{intri} + \alpha L_{integ} + \beta L_{recon} + \gamma L_{dyna} + \beta L_{consis}), \quad (14)$$

with $\alpha$, $\beta$, and $\gamma$ as adjustable weights to balance the magnitude of each term, ensuring that no single term dominates during training.

During inference, given the moment $t_s$, the model uses the state from the nearest observation moment $t_{obs}$ as an initial value, integrating over the time interval $[t_{obs}, t_s]$ in Koopman space to give the generalized model state $\theta_s^{obs \to s}$ at the desired test moment.

**Analysis and Control of the Temporal Domain Generalization.** Integrating Koopman theory into continuous temporal domain modeling facilitates the application of optimization, estimation, and control techniques, particularly through the spectral properties of the Koopman operator. We remark that the Koopman operator serves as a pivotal interface for analyzing and controlling the generalization process. Constraints imposed on the Koopman space will be equivalently mapped to the Model space. For instance, the eigenvalues of $\mathcal{K}$ are crucial as they provide insights into the system's stability and dynamics, as illustrated below:

$$z_i^{j \to i} = z_j + \int_{t_j}^{t_i} \mathcal{K} z_\tau \, d\tau, \quad \text{where } \lambda_i \text{ is an eigenvalue of } \mathcal{K} \quad (15)$$

1. **System Assessment.** The generalization process is considered stable if all $\lambda_i$ satisfy $\text{Re}(\lambda_i) < 0$. Conversely, the presence of any $\lambda_i$ such that $\text{Re}(\lambda_i) > 0$ indicates instability in the system. When $\text{Re}(\lambda_i) = 0$, the system may exhibit oscillatory behavior. By analyzing the locations of these poles in the complex plane, we can assess the system's long-term dynamics, helping us identify whether the generalized model is likely to collapse in the future.

2. **Behavioral Constraints.** Adding explicit constraints to $\mathcal{K}$ can guide the generalization toward desired behaviors. This process not only facilitates the incorporation of prior knowledge about domains but also tailors the system to specific characteristics. To name just a few, if the data dynamics are known to be periodic, configuring $\mathcal{K}$ as $\mathcal{K} = B - B^T$, with $B$ as learnable parameters, ensures that the model system exhibits consistent periodic behavior since the eigenvalues of $B - B^T$ are purely imaginary values. Additionally, employing a low-rank approximation such as $\mathcal{K} = UV^T$, with $U, V \in \mathbb{R}^{n \times k}$ and $k < n$, allows the model to concentrate on the most significant dynamical features and explicitly control the degrees of freedom of the generalization process.

**Theoretical Analysis.** In this work, we theoretically proved that our proposed continuous-time TDG method has a smaller or equal error compared to the discrete-time method for approximating temporal distribution drift. This demonstrates that *the ODE method provides a more accurate approximation due to its consideration of the integral of changes over time, reducing the accumulation of errors compared to the step-wise updates of the discrete-time methods.*

**Theorem 2** (Superiority of continuous-time methods over discrete-time methods (informal))**.** *Continuous-time methods have smaller or equal errors compared to discrete-time methods in approximating temporal distribution drift, due to its consideration of the integral of changes over time.*

The formal version and proof of Theorem 2 are given in Appendix C. We also provide a detailed model complexity analysis in Appendix A.1.4.

## 5 Experiment

In this section, we present the performance of the Koodos in comparison to other approaches through both quantitative and qualitative analyses. Our evaluation aims at *1) assessing the generalizability of Koodos in continuous temporal domains; 2) assessing whether Koodos captures the correct underlying data dynamics*; and *3) assessing whether Koodos can use inductive bias to guide the behavior of the generalization*. Detailed experiment settings (i.e., dataset details, baseline details, hyperparameter settings, ablation study, scalability analysis, and sensitivity analysis) are demonstrated in Appendix A.1. Besides, since the TDG is a special case of the CTDG, we also conducted experiments on the traditional discrete temporal domain generalization task. Results are shown in Appendix B.

Table 1: Performance comparison on continuous temporal domain datasets. The classification tasks report Error rates (%) except for the AUC for the Twitter dataset. The regression tasks report MAE. 'N/A' implies that the method does not support the task.

| Model | Classification | | | | Regression | |
|---|---|---|---|---|---|---|
| | **2-Moons** | **Rot-MNIST** | **Twitter** | **Yearbook** | **Cyclone** | **House** |
| **Offline** | $13.5 \pm 0.3$ | $6.6 \pm 0.2$ | $0.54 \pm 0.09$ | $8.6 \pm 1.0$ | $18.7 \pm 1.4$ | $19.9 \pm 0.1$ |
| **LastDomain** | $55.7 \pm 0.5$ | $74.2 \pm 0.9$ | $0.54 \pm 0.12$ | $11.3 \pm 1.3$ | $22.3 \pm 0.7$ | $20.6 \pm 0.7$ |
| **IncFinetune** | $51.9 \pm 0.7$ | $57.1 \pm 1.4$ | $0.52 \pm 0.01$ | $11.0 \pm 0.8$ | $19.9 \pm 0.7$ | $20.6 \pm 0.2$ |
| **IRM** | $15.6 \pm 0.2$ | $8.6 \pm 0.4$ | $0.53 \pm 0.11$ | $8.3 \pm 0.5$ | $18.0 \pm 0.8$ | $19.8 \pm 0.2$ |
| **V-REx** | $12.8 \pm 0.2$ | $8.6 \pm 0.3$ | $0.58 \pm 0.05$ | $8.9 \pm 0.5$ | $17.7 \pm 0.5$ | $20.2 \pm 0.1$ |
| **CIDA** | $18.7 \pm 2.0$ | $8.3 \pm 0.7$ | $0.63 \pm 0.03$ | $8.4 \pm 0.8$ | $17.0 \pm 0.4$ | $10.2 \pm 1.0$ |
| **TKNets** | $39.6 \pm 1.2$ | $37.7 \pm 2.0$ | $0.57 \pm 0.04$ | $8.4 \pm 0.3$ | N/A | N/A |
| **DRAIN** | $53.2 \pm 0.9$ | $59.1 \pm 2.3$ | $0.57 \pm 0.04$ | $10.5 \pm 1.0$ | $23.6 \pm 0.5$ | $9.8 \pm 0.1$ |
| **DRAIN-$\Delta t$** | $46.2 \pm 0.8$ | $57.2 \pm 1.8$ | $0.59 \pm 0.02$ | $11.0 \pm 1.2$ | $26.2 \pm 4.6$ | $9.9 \pm 0.1$ |
| **DeepODE** | $17.8 \pm 5.6$ | $48.6 \pm 3.2$ | $0.64 \pm 0.02$ | $13.0 \pm 2.1$ | $18.5 \pm 3.3$ | $10.7 \pm 0.4$ |
| **Koodos (Ours)** | $\mathbf{2.8 \pm 0.7}$ | $\mathbf{4.6 \pm 0.1}$ | $\mathbf{0.71 \pm 0.02}$ | $\mathbf{6.6 \pm 1.3}$ | $\mathbf{16.4 \pm 0.3}$ | $\mathbf{9.0 \pm 0.2}$ |

**Datasets.** We compare with classification datasets: Rotated Moons (2-Moons), Rotated MNIST (Rot-MNIST), Twitter Influenza Risk (Twitter), and Yearbook; and the regression datasets: Tropical Cyclone Intensity (Cyclone), House Prices (House). More details can be found in Appendix A.1.1.

**Baselines.** We employ three categories of baselines: **Practical baselines**, including 1) Offline; 2) LastDomain; 3) IncFinetune; 4) IRM [1]; 5) V-REx [25]. **Discrete temporal domain generalization methods**, including 1) CIDA [48]; 2) TKNets [57]; 3) DRAIN [2]; 4) DRAIN-$\Delta t$. **Continuous temporal domain generalization methods**, including 1) DeepODE [11]. Comparison method details can be found in Appendix A.1.2.

**Metrics.** Error rate (%) is used for classification tasks. As the Twitter dataset has imbalanced labels, the Area Under the Curve (AUC) of the Receiver Operating Characteristic (ROC) curve is used. Mean Absolute Error (MAE) is used for regression tasks. All models were trained on training domains and then deployed on all unseen test domains. Each method's experiments were repeated five times, with mean results and standard deviations reported. Detailed parameter settings for each dataset are provided in Appendix A.1.3.

### 5.1 Quantitative analysis: generalization across continuous temporal domains

We first present the performance of our proposed method against baseline methods, highlighting results from Table 1. Koodos exhibits outstanding generalizability across continuous temporal domains. A key observation is that all baseline models struggle to handle synthetic datasets, particularly challenged by the continuous and substantial concept drift (i.e., 18 degrees of rotation per second). In real-world datasets, methods like CIDA, DRAIN, and DeepODE demonstrate effectiveness in certain cases. However, the performance gap between them and Koodos highlights the importance of explicitly considering continuous temporal modeling. For instance, while DRAIN attempts to address domain dynamics through a probabilistic sequential approach via LSTM, this introduces considerable errors due to the inherent discrete temporal. Moreover, while DRAIN-$\Delta t$ and DeepODE adjust for temporal irregularities accordingly, they fail to adequately synchronize the data and model dynamics, leading to unsatisfactory results. In contrast, Koodos establishes a promising approach and a benchmark in CTDG tasks, with quantitative analysis firmly confirming the approach's superiority.

### 5.2 Qualitative analysis: data dynamics and the learned model dynamics

We conducted a qualitative comparison of different models by visualizing their decision boundaries on the 2-Moons dataset, as depicted in Fig. 3. Each row represents a different method: DRAIN-$\Delta t$, DeepODE, and Koodos, with the timeline at the bottom tracking progress through test domains. DRAIN-$\Delta t$ displays the highest error rate, showing substantial deviation from the anticipated trajectories, especially after the third domain. We also observe that DRAIN-$\Delta t$ seems to freeze when predicting multiple steps, likely due to its underlying model, DRAIN, uses a recursive training strategy within a single domain and is explicitly designed for one-step prediction. DeepODE shows a relatively better performance. It benefits from leveraging ODEs to maintain continuity in the model dynamics. However, the nonlinear variation of the predictive model parameters complicates its ability to abstract and simplify the real dynamics. Its predictions start close to the desired paths but diverge over time. Finally, Koodos exhibits the highest performance with clear and concise boundaries,

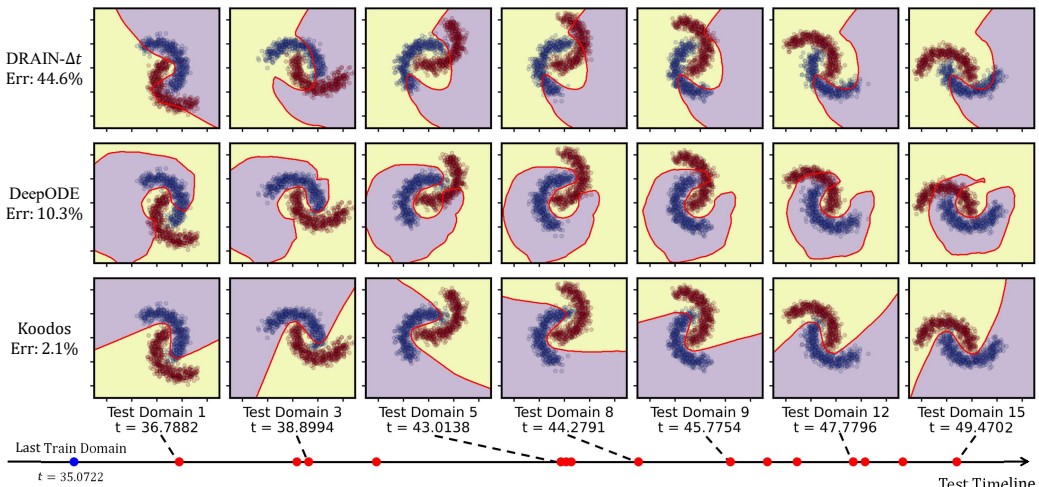

Figure 3: Visualization of decision boundary of the 2-Moons dataset (purple and yellow show data regions, red line shows the decision boundary). Top to bottom compares two baseline methods with ours; left to right shows partial test domains (all test domains are marked with red points on the timeline). All models are learned using data before the last train domain.

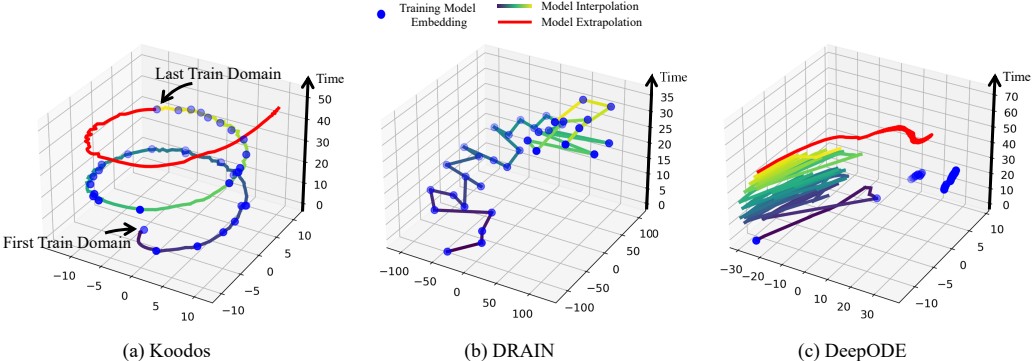

Figure 4: Interpolated and extrapolated predictive model trajectories. **Left**: Koodos captures the essence of generalization through the harmonious synchronization of model and data dynamics; **Middle**: DRAIN, as a probabilistic model, fails to capture continuous dynamics, which is presented as jumps from one random state to another. **Right**: DeepODE demonstrates a certain degree of continuity, but the dynamics are incorrect.

consistently aligning with the actual dynamics and maintaining high fidelity across all tested domains, showcasing its robustness in continuous temporal domain modeling and generalization.

Fig. 4(a) demonstrates the space-time evolution of the generalization process of Koodos. By applying t-SNE, the predictive model parameters are reduced to a 2-dimensional representation space, plotted against the time on the Z-axis. We used Koodos to interpolate the model states (35 seconds displayed in blue-yellow line) among the training domain states (marked by blue docs) in steps of 0.2 seconds, and similarly extrapolated 75 steps (15 seconds displayed in red line). The visualization clearly shows that Koodos synchronizes the model dynamics with the data dynamics: the interpolation creates a cohesive, upward-spiraling trajectory transitioning from the first to the last training domain, while the extrapolation correctly extends this trajectory outward into a new area, demonstrating the effective of Koodos from another intuitive way. We also show the space-time evolution of baseline models in Fig. 4(b,c), in which we do not find meaningful patterns of the generalization process.

## 5.3 Analysis and control of the generalization process

The learned Koopman operator provides valuable insights into the dynamics of generalized predictive models. We analyze the behavior of the learned Koodos model on the 2-Moons dataset, focusing on the eigenvalues of the Koopman operator. As illustrated in Fig. 5(a), the eigenvalues are distributed

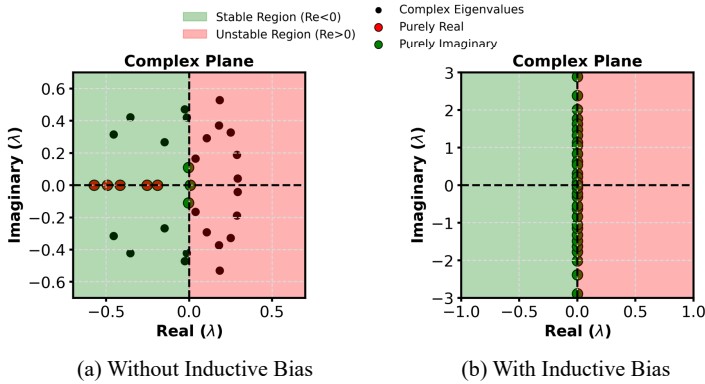

(a) Without Inductive Bias   (b) With Inductive Bias

Figure 5: Eigenvalue distribution of the Koopman operator. **Left**: $\mathcal{K}$ as learnable; **Right**: $\mathcal{K} = B - B^T$ with $B$ as learnable.

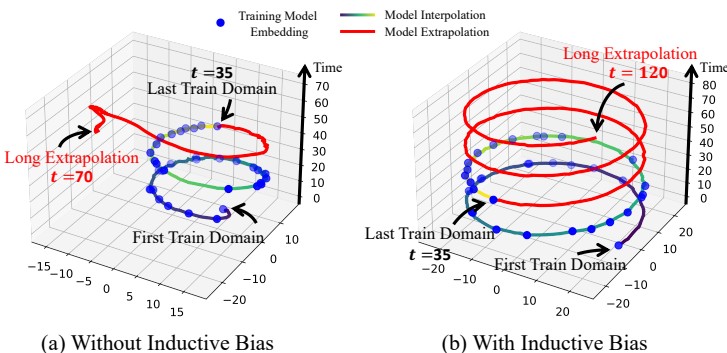

(a) Without Inductive Bias   (b) With Inductive Bias

Figure 6: Extremely long-term extrapolated predictive model trajectories in uncontrolled and controlled settings. **Left**: $\mathcal{K}$ as learnable; **Right**: $\mathcal{K} = B - B^T$ with $B$ as learnable.

across both stable (Re<0) and unstable (Re>0) regions on the complex plane. The spectral distribution suggests that while Koodos performs effectively across all tested domains, it will eventually diverge towards instability and finally collapse. To validate, we extended the extrapolation of the Koodos to an extremely long-term (i.e., 35 seconds future). Results depicted in Fig. 6(a) demonstrate that the generalized model's trajectory significantly deviates from the anticipated spiral path, suggesting that extremely long-term generalization will end up with the accumulation of errors.

Fortunately, Koodos's innovative design allows it to incorporate knowledge that extends beyond the observable data. By configuring the Koopman operator as $\mathcal{K} = B - B^T$, we ensure that all eigenvalues of the final learned $\mathcal{K}$ are purely imaginary (termed Koodos$^+$), promoting stable and periodic behavior. This adjustment is reflected in Fig. 5(b), where the eigenvalues are tightly clustered around the imaginary axis. As shown in Fig. 6(b), the embeddings and trajectories of Koodos$^+$ are cohesive and maintain stability over extended periods. Remarkably, even with 85 seconds of extrapolation, Koodos$^+$ shows no signs of performance degradation, highlighting the significance of human inductive bias in improving the robustness and reliability of the generalization process.

## 6 Conclusion

We tackle the problem of continuous temporal domain generalization by proposing a continuous dynamic system network, Koodos. We characterize the dynamics of the data and determine its impacts on model dynamics. The Koopman operator is further learned to represent the underlying dynamics. We also design a comprehensive optimization framework equipped with analysis and control tools. Extensive experiments show the efficiency of our design.

## Acknowledgments and Disclosure of Funding

This work was supported by Japan Science and Technology Agency (JST) SICORP, Grant Number JPMJSC2104; JST SPRING, Grant Number JPMJSP2108; the SPRING GX Overseas Dispatch Program and the Shibasaki Scholarship Foundation.

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

# A Appendix

## A.1 Experimental details

### A.1.1 Dataset details

In this paper, we explore a variety of datasets to analyze the performance of machine learning models under conditions of continuous temporal domain. We employ the following datasets:

1. **Rotated 2-Moons** This dataset is a variant of the classic 2-entangled moons dataset, where the lower and upper moon-shaped clusters are labeled 0 and 1, and each contains 500 instances. We randomly generated 50 distinct timestamps from the continuous real number time interval $[0, 50]$. Each timestamp corresponds to a specific domain, with each domain created by rotating the moons $18°$ counterclockwise per unit of time. *The continuous concept drift is represented by the progressive positional changes of the moon-shaped clusters.*

2. **Rotated MNIST** This dataset is a variant of the classic MNIST dataset [12], comprising images of handwritten digits from 0 to 9. We randomly generated 50 distinct timestamps from the continuous real number time interval $[0, 50]$. Each timestamp corresponds to a specific domain, within which we randomly selected 1,000 images from the MNIST dataset and rotated them $18°$ counterclockwise per unit of time. *Similar to 2-Moons, the continuous concept drift means the rotation of the images.*

3. **Twitter** The Twitter dataset [58; 3] utilizes tweet data to predict flu intensity. We randomly collected streaming tweets starting at 50 arbitrary timestamps lasting 7 days during the 2010-2014 flu seasons and then examined the volume of disease-related terms to predict current flu trends. The result is validated against the Influenza-Like Illness (ILI) activity levels reported by the Centers for Disease Control and Prevention (CDC). *The continuous concept drift in this dataset is characterized by the fluctuations in tweet volumes over the years and the variation in flu activity throughout the season.*

4. **Yearbook** The Yearbook dataset [54] consists of frontal-facing yearbook portraits collected between 1930 and 2013 from 128 high schools. The task is to classify gender from images. We randomly sampled 40 years of data from the 84-year dataset, with each year representing a domain to represent the incomplete temporal domain collection process, which resulted in variable time intervals between consecutive domains. *The Yearbook dataset provides a visual record of evolving fashion styles and social norms across the decades.*

5. **Cyclone** The Cyclone dataset [10] is collected by satellite remote sensing and is dedicated to the task of tropical cyclone imagery to wind intensity. When each cyclone occurs, the satellite collects a series of images for its entire life cycle as a domain, with the date of its occurrence representing a temporal domain time. We focused on cyclone data from the West Pacific region covering 2014 to 2016 and formed 72 continuous domains. *The dataset is event-triggered, and the variation in wind strength associated with the seasonal dates results in a continuous concept drift.*

6. **House** This dataset comprises housing price data from 2013 to 2019 and is utilized for a regression task to predict the price of a house given the features. We extracted sales data for one-month durations across 40 arbitrary time periods from 2013 through 2019. *The concept drift in this dataset is how the housing price changes over time for a certain region.*

To more clearly illustrate the temporal arbitrariness of the continuous temporal domain, we plot the time distribution of all domains for each dataset in Fig. 7, where the blue line represents the timestamp of one specific domain. We use the last 30% domain of each dataset as test domains, which are marked with gray shading.

### A.1.2 Comparison methods

1. **Practical Baseline** *(1) Offline Model*: This model operates without temporal variations and is trained using Empirical Risk Minimization (ERM) across all source domains. *(2) LastDomain*: This model is also trained without temporal variations but focuses solely on the last source domain using ERM. *(3) IncFinetune*: This approach begins by applying the baseline method at the initial time point and subsequently fine-tunes the model at each following time point using a lower learning rate. *(4) IRM*: IRM is to train a predictive model

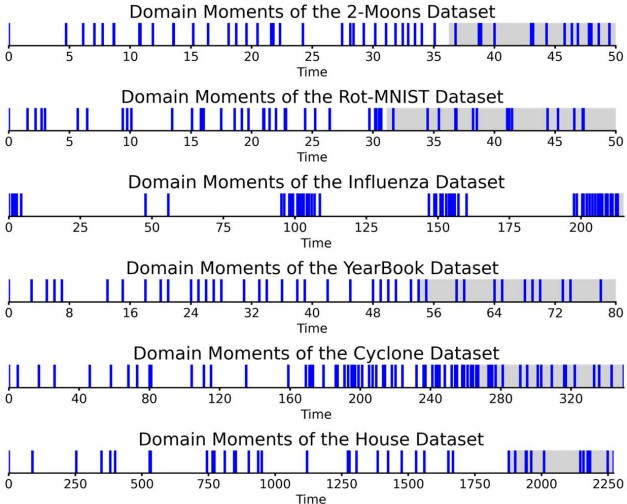

Figure 7: The distribution of continuous temporal domain datasets along the time dimension. The test domains are marked with gray shading.

over all distributions that do not rely on spurious correlations as much as possible. *(5) V-REx*: Like IRM, V-REx performs causal identification, while also providing some robustness to changes in the covariate shift by risk extrapolation.

2. **Discrete Temporal Domain Generalization Baseline** Continuous temporal domain tasks require consideration of domains for the long future rather than a limited number of steps. Existing methods are mainly unable to deal with this problem. Among them, although DRAIN [2] is designed only for predicting one-step future prediction, its internal LSTM structure has the potential of long-term future prediction, so we implemented DRAIN as a discrete baseline. We implemented two versions of it: *(1) DRAIN*: This approach tackles the temporal domain generalization problem by proposing a dynamic neural network, which uses an LSTM to capture the evolving pattern of a neural network, during the test, DRAIN will consider all future test domains as one domain, and utilizes the LSTM to predict the future parameters of this target domain. *(2) DRAIN-$\Delta t$*: A variant of DRAIN, in which the internal LSTM is changed to a continues recurrent units [44] to model irregular distributed domains. Under this setting, DRAIN-$\Delta t$ treats the test domains as an irregular time series and predicts the model state at that given moment. We also added CIDA [48] and TKNets [57] as comparisons. *(3) CIDA* treats the domain index as a continuous variable to help the discriminator using a distance-based loss. *(4) TKNets* trains a prediction model for the target domain by leveraging the evolving pattern among domains.

3. **Continuous Temporal Domain Generalization Baseline** There is currently no method to generalize the model over the continuous time domain. However, we constructed a method that can infer continuous states using a neural dynamical system. *(1) DeepODE*: We train a deep neural Ordinary Differential Equations (NeuralODEs) [11] to model the continuous dynamics of the predictive model parameters. DeepODE consists of three structures. It first uses an encoder to embed the model parameters as dense vectors, then uses NeuralODEs to infer their state in the desired future, and later uses a decoder to transfer the state back to the parameters. DeepODE treats the first domain as an initial value and infers the state of future training domains at once, using the inferred model's performance on training domains to optimize its autoencoder and NeuralODEs. In the testing phase, when a specific moment is given, we use DeepODE to infer the prediction model parameters at that moment, and then reload the estimated parameters into the prediction model for prediction.

### A.1.3 Model configuration

We use autoencoder to achieve the $\varphi$ and $\varphi^{-1}$, which we call Encoder and Decoder in the following. A linear transformation layer without bias (named Dynamic Model) is served as the Koopman operator. In certain complex data dynamics, the Koopman operator needs to expand its dimensions (potentially up to infinity) for a better approximation of the dynamics. Since it's not feasible to increase the

dimensions endlessly, we approximate the infinitely wide Koopman operator using multiple linear layers with ReLU. In practice, we found that two linear layers work well enough. Additionally, Predictive Models for each domain are trained to provide initial parameter values for Generalized Models. We use NeuralODEs [11] as the ODEs solvers. For large models with massive parameters, we treat the head network as a feature extractor shared by all domains, inferring only the state of the remaining layers. All experiments are conducted on a 64-bit machine with two 20-core Intel Xeon Silver 4210R CPU @ 2.40GHz, 378GB memory, and four NVIDIA GeForce RTX 3090. We use Adam Optimizer for all experiments, and we specify the architecture as well as other details for each dataset as follows:

1. **Rotated 2-Moons** The Predictive Model consists of 3 hidden layers, with a dimension of 50 each. We use a ReLU layer after each layer and a Sigmoid layer after the output layer. The Encoder and Decoder are both a 4-layer Multi-Layer Perceptron (MLP), with dimensions of $[1024, 512, 128, 32]$ for each layer. A 32-dimensional linear layer Dynamic Model serves as the Koopman operator. The learning rate for the Predictive Model is set at $1 \times 10^{-2}$, while for the others is set at $1 \times 10^{-3}$.

2. **Rotated MNIST** The Prediction Model features a convolutional neural network architecture comprising three convolutional layers with channel $[32, 32, 64]$ and a kernel size of 3. Each convolutional layer is followed by a ReLU layer and a max pooling layer with a kernel size of 2. Following the flattening, the network includes two linear layers with dimensions $[128, 10]$. A dropout layer is added between the linear layers to prevent overfitting. We treat the convolutional layer as the feature extractor. The Encoder and Decoder are both a 4-layer MLP with dimensions of $[1024, 512, 128, 32]$. A 32-dimensional two linear layers Dynamic Model serves as the Koopman operator. The learning rate for all parts is set at $1 \times 10^{-3}$.

3. **Twitter** The Prediction Model consists of 3 hidden layers, with a dimension of $[128, 32, 1]$. We use the ReLU layer after each layer and a Sigmoid layer after the output layer. The Encoder and Decoder are both a 4-layer MLP, with dimensions of $[1024, 512, 128, 32]$ for each layer. A 32-dimensional two linear layers Dynamic Model serves as the Koopman operator. The learning rate for all parts is set at $1 \times 10^{-3}$.

4. **Yearbook** The Prediction Model features a convolutional neural network architecture comprising three convolutional layers with channel $[32, 32, 64]$ and a kernel size of 3. Each convolutional layer is followed by a ReLU layer and a max pooling layer with a kernel size of 2. Following the flattening, the network includes three hidden linear layers with dimensions $[128, 32, 1]$. A dropout layer is added between the linear layers to prevent overfitting. The convolutional layer is used as the feature extractor. The Encoder and Decoder are both a 4-layer MLP with dimensions of $[1024, 512, 128, 32]$. A 32-dimensional two linear layers Dynamic Model serves as the Koopman operator. The learning rate is set at $1 \times 10^{-3}$.

5. **Cyclone** The Prediction Model features a convolutional neural network architecture comprising four convolutional layers with channel $[32, 32, 64, 64]$ and a kernel size of 3. Each convolutional layer is followed by a ReLU layer and a max pooling layer with a kernel size of 2. Following the flattening, the network includes three hidden linear layers with dimensions $[128, 32, 1]$. We treat the convolutional layer as the feature extractor. The Encoder and Decoder are both a 4-layer MLP, with dimensions of $[1024, 512, 128, 32]$ for each layer. A 32-dimensional 2 linear layers Dynamic Model serves as the Koopman operator. The learning rate for all parts is set at $1 \times 10^{-3}$.

6. **House** The Prediction Model consists of 3 hidden layers with a dimension of 400. We use the ReLU layer after each layer. The Encoder and Decoder are both a 4-layer MLP with dimensions of $[1024, 512, 128, 32]$. A 32-dimensional two linear layers Dynamic Model serves as the Koopman operator. The learning rate for all parts is set at $1 \times 10^{-3}$.

### A.1.4   Complexity analysis

In our framework, coordinate transformations between spaces are implemented using an autoencoder achieved by MLP. The complexity of these transformations is described by a linear relationship in terms of $N$, expressed as $\mathcal{O}(2(Nd + E) + F)$, where $N$ denotes the number of parameters $\theta$ in the predictive models, $d$ denotes the number of neurons in the first MLP layer, $E$ denotes for the number of parameters remaining in the autoencoder, and $F$ denotes the number of parameters in the Koopman operator. In many deep learning tasks, significant portions of the model's layers function as representation learning and can be shared, resulting in a small $N$ and controlled model complexity.

Table 2: Ablation test results for different datasets.

| Ablation | 2-Moons | Rot-MNIST | Cyclone |
|---|---|---|---|
| ✗$L_{integ}$ | $27.4 \pm 20.3$ | $42.7 \pm 1.0$ | $55.0 \pm 1.0$ |
| ✗$L_{recon}$ | $3.2 \pm 0.1$ | $6.3 \pm 0.3$ | $17.2 \pm 0.4$ |
| ✗$L_{dyna}$ | $5.9 \pm 5.1$ | $31.7 \pm 28.9$ | $23.3 \pm 4.9$ |
| ✗$L_{consis}$ | $30.5 \pm 18.4$ | $5.1 \pm 0.4$ | $17.4 \pm 0.5$ |
| ✗$K.S$ | $37.2 \pm 7$ | OOM | OOM |
| Koodos | $\mathbf{2.8 \pm 0.7}$ | $\mathbf{4.6 \pm 0.1}$ | $\mathbf{16.4 \pm 0.3}$ |

### A.1.5 Ablation study

We conducted an ablation study to systematically assess the contribution of various constraints and components within Koodos on two classification datasets: 2-Moons, Rot-MNIST, and a regression dataset Cyclone. The results are summarized in the Table 2. For each test, specific loss constraints were removed, indicated by ✗, and we also tested the effect of bypassing the Koopman Space but learning dynamics directly in the parameter space, designated as ✗K.S.

As shown in the table, each component of constraints contributes to robust performance across all tested datasets. The removal of particular elements has marked effects on model stability and accuracy. Specifically, the absence of $L_{integ}$ and $L_{dyna}$ constraints leads to significant performance degradation, highlighting their essential roles in learning the correct dynamics of the model through continuous temporal domain data. Removing other constraints leads to unpredictable results on certain datasets or a decline in effectiveness, indicating their importance in ensuring system effectiveness and stability under various input conditions. The considerable increase in error observed with the ✗K.S on the 2-Moons dataset demonstrates the challenges associated with modeling nonlinear dynamics directly in the parameter space, and the out-of-memory (OOM) problems with Rot-MNIST and Cyclone upon the removal of the Koopman Space suggest that this component is not only pivotal but also computationally demanding.

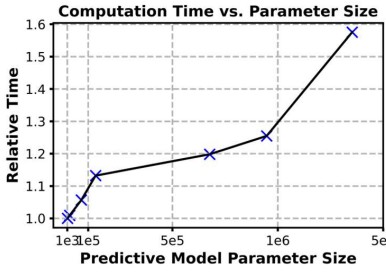
(a) Scalability for Parameter Size

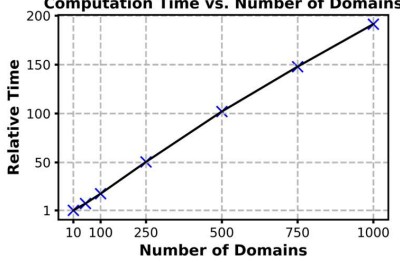
(b) Scalability for Domain Number

Figure 8: Scalability analysis w.r.t the number of parameters and the number of domains.

Table 3: Cost of training and testing time.

| Model | Train Time (s) | Test Time (s) |
|---|---|---|
| DRAIN | 67 | <1 |
| DeepODE | 289 | 1 |
| CIDA | 610 | 3.05 |
| TKNets | 948 | 2 |
| Koodos | 566 | 2.83 |

### A.1.6 Scalability analysis

We conducted a comprehensive analysis to evaluate the scalability of our system concerning both the number of parameters of the predictive model and the number of domains, as shown in Fig. 8. Computational times are normalized relative to the shortest time to provide a consistent basis.

To explore how the size of predictive model parameters affects runtime, we experimented with various configurations by varying the depth and number of neurons in the model. These configurations were tested on the Cyclone dataset with parameter counts ranging from 2,000 to over one million. As depicted in Fig. 8(a), there is a relatively linear and low growth rate in computation time as the parameter size increases. This gradual increase suggests that the autoencoder in Koodos to encode the parameter space into a Koopman space effectively mitigates the impact of increased parameter scale, maintaining computational efficiency even as predictive model complexity grows.

To assess the effect of domain count on runtime, we generated synthetic 2-Moons datasets with varying domain numbers: 10, 50, 100, 250, 500, 750, and 1000 domains, each with 200 training instances categorized into two labels. The runtime is plotted against the number of domains in Fig. 8(b), demonstrating a linear increase in computational time. This linear progression aligns with expectations for a system processing a sequence of input domains through ODEs, indicating predictable and manageable scalability as domain count increases.

We further conduct the model scalability analysis by comparing the running time of Koodos with other state-of-the-art baselines: DRAIN, DeepODE, CIDA, and TKNets in the 2-Moons dataset. As shown in Table 3, our model strikes a good balance between training time and effectiveness.

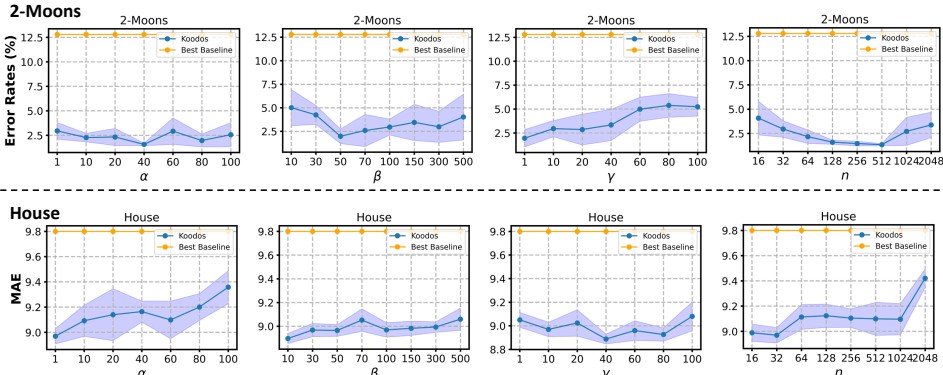

Figure 9: Sensitivity analysis.

### A.1.7 Sensitivity analysis

We conducted the sensitivity analysis on 2-Moons and House datasets to understand the hyperparameters in Koodos: loss weights $(\alpha, \beta, \gamma)$ and the dimensions $n$ of the Koopman operator $\mathcal{K}$. The loss weights are set based on the magnitude of each loss term. For instance, in the 2-Moons dataset, the cross-entropy loss $L_{intri}$ and $L_{integ}$ are approximately 1 after convergence, the $L_{recon}$ and $L_{consis}$ in the Model space are around 0.01, and $L_{dyna}$ in the Koopman space is around 0.1. Accordingly, we set the initial values of $\alpha, \beta, \gamma$ to 1, 100, and 10, respectively. We then adjust each term independently within its respective range: $\alpha$ and $\gamma$ are varied from 1 to 100, and $\beta$ from 10 to 1000. The dimension $n$ of the Koopman operator varies from 16 to 2048. Fig. 9 shows the results.

It can be seen that: (1) Koodos exhibits stable behavior with controlled variance across a wide range of hyperparameter values, evidencing its insensitivity to hyperparameter variations. (2) Setting the loss weights of $(\alpha, \beta, \gamma)$ by the magnitude of each loss term is sufficient for achieving good performance, and fine-tuning the weights may give better results. (3) A hundreds-dimension approximation of the Koopman operator is sufficient to achieve good results. Too high may lead to overfitting.

### A.1.8 Limitations

The limitation arises from the assumptions inherent in the research area of the temporal domain generalization framework. TDG assumes that domain pattern drifts follow certain predictable, smooth patterns, allowing for modeling future changes as a sequence. CTDG extends TDG to continuous time, further generalizing the problem to handle domains collected at arbitrary times. Thus, their assumptions are aligned in focusing on smooth, predictable concept drift. To address other kinds of drift, alternative frameworks may be required. However, different frameworks can be combined to address the full spectrum of domain generalization challenges. Exploring such a comprehensive approach may represent a promising research direction.

Table 4: Performance comparison on discrete temporal domain datasets. The classification tasks report Error rates (%), and the regression tasks report MAE.

| Model | Classification | | | | Regression | |
|---|---|---|---|---|---|---|
| | **2-Moons** | **Rot-MNIST** | **ONP** | **Shuttle** | **House** | **Appliance** |
| **Offline** | $22.4 \pm 4.6$ | $18.6 \pm 4.0$ | $33.8 \pm 0.6$ | $0.77 \pm 0.1$ | $11.0 \pm 0.36$ | $10.2 \pm 1.1$ |
| **LastDomain** | $14.9 \pm 0.9$ | $17.2 \pm 3.1$ | $36.0 \pm 0.2$ | $0.91 \pm 0.18$ | $10.3 \pm 0.16$ | $9.1 \pm 0.7$ |
| **IncFinetune** | $16.7 \pm 3.4$ | $10.1 \pm 0.8$ | $34.0 \pm 0.3$ | $0.83 \pm 0.07$ | $9.7 \pm 0.01$ | $8.9 \pm 0.5$ |
| **CDOT** | $9.3 \pm 1.0$ | $14.2 \pm 1.0$ | $34.1 \pm 0.0$ | $0.94 \pm 0.17$ | - | - |
| **CIDA** | $10.8 \pm 1.6$ | $9.3 \pm 0.7$ | $34.7 \pm 0.6$ | - | $9.7 \pm 0.06$ | $8.7 \pm 0.2$ |
| **GI** | $3.5 \pm 1.4$ | $7.7 \pm 1.3$ | $36.4 \pm 0.8$ | $0.29 \pm 0.05$ | $9.6 \pm 0.02$ | $8.2 \pm 0.6$ |
| **DRAIN** | $3.2 \pm 1.2$ | $7.5 \pm 1.1$ | $38.3 \pm 1.2$ | $0.26 \pm 0.05$ | $9.3 \pm 0.14$ | $6.4 \pm 0.4$ |
| **Koodos (Ours)** | $\mathbf{1.3 \pm 0.4}$ | $\mathbf{7.0 \pm 0.3}$ | $\mathbf{33.5 \pm 0.4}$ | $\mathbf{0.24 \pm 0.04}$ | $\mathbf{8.8 \pm 0.19}$ | $\mathbf{4.8 \pm 0.3}$ |

## B  Experiments on discrete temporal domain generalization

While our primary focus is on Continuous Temporal Domain Generalization (CTDG), we have also conducted experiments in the more conventional context of Temporal Domain Generalization (TDG). In essence, TDG can be seen as a specialized case of CTDG, where making the continuous system with a step size fixed at 1. Such a setting does not affect the proper ability of the CTDG model to capture the data dynamics and the corresponding model dynamics.

In TDG experiments, we follow the problem definition in DRAIN [2]. Formally, a sequence of data domains is collected over discrete time intervals. Each domain, denoted as $\mathcal{D}_t$, corresponds to a dataset collected at a specific time $t$ where $t = 1, 2, ..., T$. We have a sequence of domains $\{\mathcal{D}_1, \mathcal{D}_2, ..., \mathcal{D}_T\}$, where each domain $\mathcal{D}_t = \{(x_i^t, y_i^t)\}_{i=1}^{N_t}$ consists of $N_t$ instances with features $x_i^t \in X_t$ and labels $y_i^t \in Y_t$, and $X_t$ and $Y_t$ represent the random variables for features and targets. The goal in TDG is to train a predictive model $g(X_t; \theta_t)$ on these historical domains, and modeling the dynamics of $\{\theta_1, \theta_2, ..., \theta_T\}$, then predict the $\theta_{T+1}$ for the unseen future domain $\mathcal{D}_{T+1}$ at time $t = T + 1$, finally do the prediction task $Y_{T+1}^* = g(X_{T+1}; \theta_{T+1})$.

We have replicated the experimental setups typically used in TDG research. Specifically, we used datasets, settings, and hyperparameters from the DRAIN [2]. We compare with the following classification datasets: Rotated Moons (2-Moons), Rotated MNIST (Rot-MNIST), Online News Popularity (ONP), and Shuttle; and the following regression datasets: House prices dataset (House), Appliances energy prediction dataset (Appliance). Note that the first two datasets differ from datasets used in our CTDG main experiments in Exp. 5, as here they are time-regularly distributed and only have 1-step future domain to predict. We keep the architecture of the predictive model for each dataset the same as DRAIN [2].

Performance comparison of all methods in terms of misclassification Error (in %) for classification tasks and mean absolute error (MAE) for regression tasks. All experiments are repeated 5 times for each method, and we report the average results and the standard deviation in the quantitative analysis.

The results are shown in Table 4. The Koodos model consistently outperforms all baselines across various datasets, demonstrating its robustness and superior adaptability also made to traditional temporal domain generalization tasks. Notably, Koodos significantly reduces error margins, especially in real-world datasets such as House and Appliance. The exceptional performance of Koodos can be attributed to its fundamental continuous dynamic system design, which effectively captures and synchronizes dynamic changes in data and models.

# C   Theorem proofs

## Assumptions

1. **Continuous Data Distribution Drift**: - Let $p(t)$ be the data distribution at time $t$. - Assume $p(t)$ is continuously differentiable, and the drift $\frac{dp(t)}{dt}$ is Lipschitz continuous with constant $L$.

2. **Training and Target Domains**: - The model is trained on a sequence of training domains $\{p(t_i)\}_{i=1}^{T}$ where $t_1 < t_2 < \ldots < t_T$. - The target domain is at a future time $s$, with the data distribution $p(s)$. - The time intervals $\Delta t_i = t_{i+1} - t_i$ can be arbitrary.

3. **Model Training**: - For the ODE model, let $x(t)$ be the state at time $t$, governed by the ODE:
$$\frac{dx(t)}{dt} = f(x(t), t),$$
where $f$ is a smooth function. - For the RNN model, let $h_{t_i}$ be the state at discrete time $t_i$, updated by:
$$h_{t_{i+1}} = h_{t_i} + \Delta t_i \cdot \phi(h_{t_i}, x_{t_i}),$$
where $\phi$ is a smooth function and $x_{t_i}$ is the input at time $t_i$.

4. **Generalization Error**: - The generalization error on the target domain $p(s)$ depends on the accumulated training errors and the error propagation from the last training domain $p(t_T)$ to the target domain $p(s)$.

**Theorem 3** (Formal version of Theorem 2). *Given the assumptions, a continuous-time method using an Ordinary Differential Equation (ODE) provides a smaller or equal generalization error on an unseen target domain compared to a discrete-time method using a Recurrent Neural Network (RNN), due to its more accurate approximation of the data distribution drift over time.*

*Proof.* Below is the formal proof:

**Accumulated Training Error**

1. **ODE Model**: - The state $x(t + \Delta t_i)$ can be approximated by the ODE integral:
$$x(t_i + \Delta t_i) = x(t_i) + \int_{t_i}^{t_i + \Delta t_i} f(x(\tau), \tau)\, d\tau. \tag{16}$$
- Using the Taylor series expansion for $f(x(t), t)$:
$$x(t_i + \Delta t_i) = x(t_i) + f(x(t_i), t_i)\Delta t_i + \frac{(\Delta t_i)^2}{2}\frac{\partial f}{\partial t} + \mathcal{O}((\Delta t_i)^3). \tag{17}$$
- The error for one step is:
$$\text{Error}_{\text{ODE, step}} = \mathcal{O}((\Delta t_i)^2). \tag{18}$$
- The accumulated error over $T$ steps is:
$$\text{Error}_{\text{ODE, train}} = \sum_{i=1}^{T-1} \mathcal{O}((\Delta t_i)^2). \tag{19}$$
- Given that the intervals $\Delta t_i$ can be large, the accumulated error becomes:
$$\text{Error}_{\text{ODE, train}} = \mathcal{O}\left(\sum_{i=1}^{T-1}(\Delta t_i)^2\right). \tag{20}$$

2. **RNN Model**: - The state $h_{t_{i+1}}$ is updated as:
$$h_{t_{i+1}} = h_{t_i} + \Delta t_i \cdot \phi(h_{t_i}, x_{t_i}). \tag{21}$$
- Assuming $h_{t_i} \approx x(t_i)$ and $\phi(h_{t_i}, x_{t_i}) \approx f(x(t_i), t_i)$, the error for one step is:
$$\text{Error}_{\text{RNN, step}} = \mathcal{O}(\Delta t_i). \tag{22}$$
- The accumulated error over $T$ steps is:
$$\text{Error}_{\text{RNN, train}} = \sum_{i=1}^{T-1} \mathcal{O}(\Delta t_i). \tag{23}$$
- Given that the intervals $\Delta t_i$ can be large, the accumulated error becomes:
$$\text{Error}_{\text{RNN, train}} = \mathcal{O}\left(\sum_{i=1}^{T-1} \Delta t_i\right). \tag{24}$$

**Error Propagation to Target Domain**

1. ODE Model: - To handle an arbitrary future time $s$, we integrate the error over many small steps from $t_T$ to $s$:

$$\text{Error}_{\text{ODE, target}} \approx \text{Error}_{\text{ODE, train}} + \int_{t_T}^{s} \mathcal{O}((\tau - t_T)^2)\, d\tau. \tag{25}$$

- Since $\tau$ ranges from $t_T$ to $s$:

$$\text{Error}_{\text{ODE, target}} \approx \text{Error}_{\text{ODE, train}} + \mathcal{O}((s - t_T)^3). \tag{26}$$

2. RNN Model: - Similarly, the error propagation to the target domain $s$ involves many small steps:

$$\text{Error}_{\text{RNN, target}} \approx \text{Error}_{\text{RNN, train}} + \sum_{k=0}^{K-1} \mathcal{O}(\Delta t_k), \tag{27}$$

- where $K$ is the number of small steps from $t_T$ to $s$ and $\Delta t_k$ are the small intervals:

$$\text{Error}_{\text{RNN, target}} \approx \text{Error}_{\text{RNN, train}} + \mathcal{O}(s - t_T). \tag{28}$$

**Generalization Error Comparison**

- For the ODE model:

$$\text{Error}_{\text{ODE, target}} \approx \mathcal{O}\left(\sum_{i=1}^{T-1} (\Delta t_i)^2\right) + \mathcal{O}((s - t_T)^3). \tag{29}$$

- For the RNN model:

$$\text{Error}_{\text{RNN, target}} \approx \mathcal{O}\left(\sum_{i=1}^{T-1} \Delta t_i\right) + \mathcal{O}(s - t_T). \tag{30}$$

- Since $(\Delta t_i)^2$ is much smaller than $\Delta t_i$ for any $\Delta t_i$, and $(s - t_T)^3$ is much smaller than $s - t_T$, we have:

$$\text{Error}_{\text{ODE, target}} \leq \text{Error}_{\text{RNN, target}}. \tag{31}$$

$\square$

