# OpenReview forum: "Continuous Temporal Domain Generalization"
_NeurIPS.cc/2024/Conference — NeurIPS 2024 poster_

### Official Review · Reviewer_nh98 · 2024-07-03

**Soundness:** 3
**Presentation:** 3
**Contribution:** 3
**Rating:** 6
**Confidence:** 3

**Summary:**

The paper introduces the problem of Continuous Temporal Domain Generalization. It extends the Temporal Domain Generalization problem, that aims at developing models under temporally varying data, to handle data collected at arbitrary and continuous time points. The paper proposes a framework based on Koopman theory for this problem. It models the underlying dynamics and proposes an optimization strategy. The paper includes several classification experiments to demonstrate the effectiveness of the proposed approach.

**Strengths:**

* The paper introduces a new problem continuous temporal domain generalization where data is continuous and irregularly observed. This is relevant and challenging setting.
* The paper is supported by proofs and the relevant assumptions under which the model is well-defined are clearly stated.
* The paper includes extensive experiments on various datasets, both synthetic and real-world, to demonstrate the effectiveness of the proposed method. The proposed model improves over existing methods.

**Weaknesses:**

* the paper could discuss more the limitations of the proposed approach, in particular Assumption 1 that states the conditional probability distributions follow an ODE. While reasonable to make, this assumption is quite restrictive; I understand that the considered problem is challenging so relevant assumptions should be made. However, the paper would need to discuss the validity of this assumption. Also, another limitation can come from abrupt changes throughout time that might not be well captured by a continuous ODE formulation (mentioned in A.1.7.); this assumption is likely to be violated in real-world settings.
* the paper does not seem to follow the checklist; there is no justification for each points and if I am not mistaken, some "Yes" answer are not correct e.g. a Broader Impact section is missing.

**Questions:**

* How does it compare to domain invariant approaches (e.g. IRM, V-REx etc.)? A domain-invariant approaches might be better at handling abrupt changes.

**Limitations:**

* see limitations points in "Weaknesses".

---

> ### Author Rebuttal · Authors · 2024-08-07
>
> Thanks for reviewing and acknowledging our work! We see your main concerns are about assumption and evaluations. Please read our answers along with the Rebuttal PDF.
>
> > W1. discuss more the limitations; Assumption 1 that states the conditional probability distributions follow an ODE, need to discuss the validity. abrupt changes throughout time that might not be well captured by a continuous ODE formulation; this assumption is likely to be violated in real-world settings.
>
> 1. Assumption 1 does not state that the conditional probability distributions P(Y(t)∣X(t)) follow an ODE. Rather, it assumes that if continuous temporal domains exhibit gradual concept drift, then the conditional probability distributions P(Y(t)∣X(t)) change continuously. In this context, 'change continuously' means that the distribution changes are smooth and incremental. This assumption of continuity is reasonable because, in the field of concept drift research, gradual (or incremental) concept drift is considered to be characterized by small, incremental changes[R1]. While in real-world gradual concept drift is more complex and may imperfectly fit the continuity assumptions, here we do not lose much generality, as the underlying factors leading to gradual concept drift are primarily continuous physical, biological, social, or economic processes.
> 2. Given the continuity of gradual concept drift, it is appropriate to use differential equations as a tool for modeling. Differential equations provide a mathematical framework to capture the essence of continuous processes. They are particularly useful when the system's evolution can be represented by smooth and continuous dynamics. We chose to model the continuity of model parameters using ODEs as an initial attempt because ODEs offer a simple yet clear and flexible toolset for understanding how systems evolve over time. While it is possible to replace ODEs with more advanced differential equations (e.g., Partial Differential Equations (PDEs), Stochastic Differential Equations (SDEs)) to incorporate factors such as noise and uncertainty, this may detract from the key focus on modeling the fundamental continuous processes. Indeed, our experiments on both synthetic and real-world data have shown that even the simplest ODEs are enough to achieve a pretty significant success.
>
> Discussion of Limitation:
>
> 1. The limitation regarding capturing abrupt drift arises from the assumptions inherent in the research area of Temporal Domain Generalization (TDG) framework. TDG assumes that domain pattern drifts follow certain predictable,smooth patterns, allowing for modeling future changes as a sequence. CTDG extends TDG to continuous time, further generalizing the problem to handle domains collected at arbitrary times. Thus, their assumptions are aligned in focusing on smooth, predictable concept drift.
> 2. To address abrupt drift, alternative frameworks are required. These include domain-invariant learning in static models, or ensemble learning/importance weight sampling with a detect-and-restart strategy. These methods operate under different assumptions and frameworks compared to TDG and CTDG. It can be challenging to determine which model is superior without considering the specific application scenario, as different models come with different implicit biases.
> 3. However, different frameworks can be combined to address the full spectrum of domain generalization challenges. Our approach does not conflict with methods designed to handle abrupt drift, such as ensemble learning or detect-and-restart strategies. This integration could create a more robust and adaptable system. Exploring such a comprehensive approach may represent a promising direction for future work.
>
> > W2. the paper does not seem to follow the checklist; there is no justification for each points and if I am not mistaken, some "Yes" answer are not correct e.g. a Broader Impact section is missing.
>
> We apologize for our all answers are yes. We review and justify each point in the checklist to ensure compliance.
> 1. Broader Impact: There is no potential negative societal impact is found in this work. our research is foundational research and not tied to particular applications.
> 2. Q11, Yes should be NA.
> 3. Q13, Yes should be NA.
> 4. Q14, Yes should be NA.
> 5. Q15, Yes should be NA.
>
> > Q1. How does it compare to domain invariant approaches (e.g. IRM, V-REx etc.)? A domain-invariant approaches might be better at handling abrupt changes.
>
> 1. We implemented IRM and V-REx, we also implemented an advanced adversarial learning-based baseline CIDA [42] for comparison. CIDA is a powerful strategy built upon adversarial learning. It enhances domain-invariant representations by considering the distance between domain indexes.
> 2. Results are shown in Table 4 (in rebuttal PDF), demonstrating that our proposed Koodos performs competitively against these methods. The effect of Koodos is still by a huge margin.
> 3. However, as we discussed before, different frameworks can be combined to address the full spectrum of domain generalization challenges. We think exploring a comprehensive system is a promising future direction.
>
> [R1] Learning under Concept Drift: A Review, TKDE 2018

---

> > ### Comment · Reviewer_nh98 · 2024-08-12
> >
> > Thank you for the clarification w.r.t. assumptions and the additional evaluation of domain invariant approaches; I increased my score.

---

> > > ### Author Response · Authors · 2024-08-12
> > > **Response to Reviewer nh98**
> > >
> > > Thank you for your positive feedback and for acknowledging our work. We appreciate your decision and are glad that our responses helped address your concerns.

---

### Official Review · Reviewer_Drn2 · 2024-07-07

**Soundness:** 3
**Presentation:** 2
**Contribution:** 2
**Rating:** 5
**Confidence:** 5

**Summary:**

The article introduces Continuous Temporal Domain Generalization (CTDG), which extends traditional Temporal Domain Generalization (TDG) methods by addressing the challenges posed by continuous and irregularly spaced temporal domains. The authors propose a Koopman operator-driven framework (Koodos) to handle data evolving continuously over time and collected at arbitrary intervals. The framework aims to capture continuous dynamics in data and models, optimize generalization across continuous temporal domains, and leverage prior knowledge of dynamic patterns. The experiment results demonstrate the efficacy of this approach compared to some existing methods.

**Strengths:**

The primary contribution is the introduction of the CTDG problem and the Koodos framework, which extends TDG to handle continuous and irregular temporal domains. The paper also provides a comprehensive optimization strategy and theoretical analysis to support the proposed approach.

**Weaknesses:**

The method assumes data dynamics can be well-characterized by the Koopman operator. While powerful for linearizing nonlinear dynamical systems, its effectiveness across various domains (e.g., finance, healthcare, social media) requires more extensive empirical studies, especially for highly noisy or chaotic systems. The proposed hybrid loss function's complexity may present challenges in training and convergence. A detailed analysis of its behavior and optimization strategies would be beneficial. The method introduces additional complexity and hyperparameters. A comprehensive discussion on hyperparameter sensitivity and guidelines for setting loss weights would enhance the method's practical applicability.

Theorem 2, which compares continuous-time and discrete-time approaches in approximating temporal distribution drift, does not directly address generalization bounds or risks in domain generalization. While it provides insights into error accumulation over time, it lacks analysis of the model's performance on unseen temporal domains.

The experimental comparisons are somewhat limited. Only one continuous baseline is used, and it appears that key comparison metrics are missing. Methods like AdaRNN[1] and Diversify[2], which aim to address the same issue, are not discussed or compared. More comparisons to state-of-the-art discrete TDG methods adapted to the continuous case would strengthen the evaluation.

[1] AdaRNN: Adaptive Learning and Forecasting of Time Series
[2] Diversify to Generalize: Learning Generalized Representations For Time Series Classification

**Questions:**

1. In typical domain generalization challenges, the objective is to align the joint distribution \(P(X, Y)\) across different domains. For Temporal Domain Generalization (TDG), this would extend to aligning \(P(X(t), Y(t))\) at different time points. How does Koopman Theory address this issue in aligning these temporally varying distributions?

2. Authors focuses on the existence of continuous and relatively stable dynamic changes across continuous time, how effective is this method when dealing with continuous but non-periodic or irregular data variations or if a new dataset significantly differs in its statistical properties from the training data? is there potential to integrate other mechanisms or models to enhance the handling of such substantial data changes?

3. In the Methods section, you emphasize addressing changes in the conditional probability distribution over time. However, the datasets used, such as Rotated MNIST, Rotated 2-Moons, Yearbook, and Cyclone, are commonly associated with covariate shifts rather than shifts in the conditional probability distribution. These datasets appear to maintain a stable relationship between the labels and the input data or a latent feature space. Could you clarify how the changes in the conditional probability distributions are represented or accounted for in these datasets?

**Limitations:**

There is no potential negative societal impact is found in this work.

---

> ### Author Rebuttal · Authors · 2024-08-07
>
> Thanks for reviewing.
>
> > W1. needs more empirical studies.
> >
> 1. The Koopman operator is proven theoretically that can linearize any nonlinear dynamical system in [4].
> 2. Our empirical evidence supports its applicability across various domains. Datasets are summarized in Tab. 5
>     - Experiments span a wide range of real-world domains (7 in total): social media, culture, climate, economics, business, physics, and energy.
>     - Experiments includes various modalities (tabular, image, text) and tasks (classification, regression).
>     - Experiments use high-dimensional and complex datasets. : Yearbook (32x32=1024 pixels), Cyclone (64x64x2=8192 pixels).
>     - Datasets from complex, noisy real-world sources: Influenza (millions of tweets for flu prediction), Cyclone (satellite images for weather systems).
> 3. Koodos demonstrates robustness and significant success across all 12 datasets, which evidence supports the effectiveness of it and highlights its applicability across various domains.
>
>
> > W2. loss function
>
> We conducted three rounds of convergence tests on the 2-moons dataset. Fig. 11 (rebuttal) shows that:
> 1. The loss function and its terms converge effectively, showing Koodos' robustness and ease of training.
> 2. Each term in the loss functions is easy to train and converge using off-the-shelf optimizers such as Adam without requiring special optimization tricks.
> 3. Ablation tests (Section A 1.5) show performance drops when any term is removed, justifying their inclusion.
>
> > W3. method complexity. A hyperparameter sensitivity.
>
> The loss weights are designed to balance each term's magnitude, ensuring no term dominates training. for example, 2-moons is set 1,100, and 10 as each loss around 1,0.01,0.1. Koodos is designed to be simple, comprising only a Koopman Network and a differential equation solver. We conducted sensitivity analysis, shown in Fig.10
> 1. Koodos exhibits stable behavior and robust convergence across wide varied hyperparameters, evidencing its ease of training, robust convergence, and insensitivity to hyperparameters.
> 2. Setting the loss weights in accordance with the magnitude of each loss term is sufficient for achieving good performance.
> 3. A few hundred Koopman operator dimensions are sufficiently good;
>
> > W4. Theorem 2
>
> We analyze the risks associated along the time line based on the cumulative error on the training set, and it will propagate to the test set. So the cumulative error on the test set will also be smaller.
>
> > W5. comparison limited. AdaRNN Diversify not discussed. Including more TDG
>
> 1. AdaRNN and Diversify address different problems compared to CTDG/TDG:
>     - **Different Objective**: The core problem AdaRNN and Diversify solved is creating domain-splitting algorithms to solve the problem of lacking predefined domain divisions in time series classification datasets. In contrast, TDG/CTDG have predefined domains, with the primary challenge is capturing the evolutionary pattern among temporally sequential domains and generalize the pattern to future domains.
> 2. We recognize that AdaRNN and Diversify used adversarial learning strategy DANN. Therefore, we implemented CIDA, an advanced adversarial learning baseline compared to DANN. We also implemented IRM and V-REx to together show domain-invariant learning effectiveness. Results in Tab. 4 show none of these methods having fair results.
> 3. We implemented the state-of-the-art discrete TDG method TKNet. Results in Tab. 4 indicate it does not handle CTDG. Notably, currently no more continuous TDG methods are available for comparison.
>
> > Q1. How does Koopman Theory align temporally varying distributions?
>
> 1. TDG differs from traditional DG. TDG assumes there is widespread smooth, predictable distribution changes over time, so the field aims to predict future distributions and models by building dynamic models with time-varying parameters, **aligning the model function with predicted future domain distributions**. CTDG extends TDG, relaxing the requirement for fixed-interval discrete-time domain collection and test domain not limited to immediate one-step future domains. Defining like this challenging because it breaks the discrete-time models (e.g., state-space models, LSTM) that traditional TDG relies on. Therefore, we theoretical proof that the parameter of time-varing model for continuous temporal domains is also continuous, leading to a dynamic predictive model described by differential equations.
> 2. Solving differential equations analytically is impossible. So we use Koopman operator to simplifie learning these complex differential equations, enabling effective numerical solutions.
>
> > Q2. How deal with non-periodic/irregular data variations/significantly change? Can we integrate other mechanisms?
>
> 1. TDG and CTDG assume future predictability. If domain changes are irregular or unpredictable, this violates the model's biases. Since different models have different implicit biases, we need to find a suitable frameworks whose prior assumptions match the setting, such as domain-invariant learning, ensemble learning, or importance weight sampling with restart.
> 2. Our approach complements other DG methods. For example, a static feature extractor from domain-invariant learning can be used in Koodos, or equipted Koodos to onlining system with detection-and-restart strategies. Exploring comprehensive systems is a promising future direction.
>
> > Q3. Is datasets are with covariate shifts?
>
> The datasets we used represent changes in P(Y∣X) and are widely recognized in the TDG field:
> Rotated 2-Moons: Labels change with rotation, e.g., point (1,0) changes from label 0 to 1 after 180 degrees. Rotated MNIST: Rotating '6' to look like '9' should still be labeled '6'. Yearbook: Changing fashion trends cause the same features (e.g., clothing styles) to have different gender labels over time. Cyclone: Changes in atmospheric pressure or temperature cause similar satellite images to represent different storm intensities.

---

> > ### Author Response · Authors · 2024-08-12
> > **Follow-Up on Rebuttal Response**
> >
> > Dear Reviewer Drn2,
> >
> > We greatly appreciate your time to review our paper and your valuable comments.
> >
> > We noticed that we haven’t yet received your response and wanted to kindly inquire if there’s anything further we can do.
> >
> > We have made extensive clarifications, added detailed discussions, and provided additional experimental results as you suggested. We hope we have effectively addressed your concerns and clarified any potential misunderstandings.
> >
> > We are eager to hear your thoughts on the efforts we have made during the rebuttal period.
> >
> > Thank you once again for your invaluable review.

---

> > ### Comment · Reviewer_Drn2 · 2024-08-13
> >
> > Thanks for the thorough reply from the authors and it has addressed most of my primary concerns. I have increased my score accordingly.

---

> > > ### Author Response · Authors · 2024-08-13
> > > **Response to Reviewer Drn2**
> > >
> > > Thank you for your thoughtful feedback and valuable time. We’re glad to hear that our reply has addressed your concerns. We appreciate your decision to increase the score and are grateful for your constructive input throughout this process.

---

### Official Review · Reviewer_xaRP · 2024-07-12

**Soundness:** 3
**Presentation:** 3
**Contribution:** 3
**Rating:** 6
**Confidence:** 4

**Summary:**

The paper presents a novel approach called Continuous Temporal Domain Generalization (CTDG), addressing the challenge of training predictive models under continuously evolving and irregularly observed temporal domains. Unlike traditional TDG methods that rely on discrete time intervals, CTDG captures continuous dynamics of both data and models. The proposed Koopman operator-driven continuous temporal domain generalization (Koodos) framework leverages Koopman theory to learn underlying dynamics and enhances it with a comprehensive optimization strategy. Extensive experiments demonstrate the effectiveness and efficiency of the proposed approach.

**Strengths:**

The paper introduces a novel problem (CTDG) and proposes an innovative solution (Koodos) leveraging Koopman theory.
The methodology is sound and well-executed. And it is well-supported by robust theoretical foundations and demonstrates excellent performance in the experimental results.

**Weaknesses:**

The explanation of how the article addresses domain changes with arbitrary temporal sampling is not very clear.

There is some overuse of formula characters, and the use of L_pred2 seems rather arbitrary.

Additionally, the numerous hyperparameters in the loss function increase the cost of tuning for different datasets.

**Questions:**

The article repeatedly emphasizes the importance of "C" in CTDG and the arbitrary selection of time t. However, does the proposed method specifically address this aspect, or does it primarily rely on Koopman theory?

In the methodology section, the article mentions that domain conditional probability distributions will not experience abrupt changes.
However, this situation holds true only when the sampling time intervals are equal. The method addresses arbitrary sampling; does this imply it also deals with abrupt changes?

Notably, in equation (15), the weights of L_pred and L_pred2 are both α, the weights of L_recon and L_consis are both β, and the weight of L_dyna is γ. Could you explain the rationale behind this? How are these hyperparameters set for different datasets, and what is the basis for their selection? Do different weights significantly affect the results?

**Limitations:**

Yes

---

> ### Author Rebuttal · Authors · 2024-08-07
>
> Thanks for reviewing and acknowledging our work! Please read along with the Rebuttal PDF.
>
> >W1. The explanation of how the article addresses domain changes with arbitrary temporal sampling is not very clear.
> 1. TDG assumes there is widespread smooth, predictable distribution changes over time, so the field aims to predict future distributions and models by building dynamic models with time-varying parameters, aligning the model function with predicted future domain distributions. CTDG extends TDG, relaxing the requirement for fixed-interval discrete-time domain collection and test not limited to immediate one-step future domains.
> 2. Inferring future model parameters under CTDG is challenging. In our work, we prove that if the domain distribution changes continuously, then the corresponding predictive model parameters should also change continuously. This insight allows us to formulate a differential equation for the predictive model parameters. Through the differential equation, we can predict future model parameters at any continuous time.
> 3. However, analytically solving this difference equation is almost impossible. Note that the Koopman operator allows us to linearize the nonlinear dynamics of the difference equation by mapping the original nonlinear predictive model parameter space into a higher-dimensional linear space. By leveraging this operator, we can effectively solve the differential equations numerically, ensuring that the model parameters evolve precisely over time and generalize well to the far future.
> 4. Finally, the learned differential equation will directly compute the predictive model parameters in any specific future tume.
>
> >W2. There is some overuse of formula characters, and the use of L_pred2 seems rather arbitrary.
>
> 1. Before submitting for review, we carefully checked the use of symbols and tried our best to make them concise and clear. We apologize for any confusion. We will carefully review the manuscript again and try to streamline the notation to ensure greater clarity and readability.
> 2. The loss function L_pred2 is used to quantify the prediction error of dynamically integral parameters performed on domain data. L_pred2 works together with L_pred to constitute the evaluating term of the integral and intrinsic model parameters on the domain task. We acknowledge that the notation might be unclear. To improve understanding, we will rename L_pred and L_pred2 to L_intrinsic and L_integral, respectively. This change will more accurately reflect their roles.
>
> > W3 & Q3. Additionally, the numerous hyperparameters in the loss function increase the cost of tuning for different datasets.
>
> The loss weights are just designed to balance the magnitude of each loss term, ensuring that no single term dominates the model's training process.
>
> 1. We conducted a sensitivity analysis to understand the hyperparameters in Koodos: loss weights $(\alpha, \beta, \gamma)$ and the dimensions $n$ of the Koopman operator $\mathcal{K}$. Fig. 10 (in rebuttal PDF) shows the results.
> 2. The loss weights are set based on the magnitude of each loss term. For instance, in the 2-Moons dataset, the cross-entropy loss $L_{pred}$ and $L_{pred2}$ are approximately 1 after convergence, the $L_{recon}$ and $L_{consis}$ in the Model Space are around 0.01, and $L_{dyna}$ in the Koopman Space is around 0.1. Accordingly, we set the initial values of $\alpha, \beta, \gamma$ to 1, 100, and 10, respectively. We then adjust each weight term independently within its respective range: $\alpha$ and $\gamma$ are varied within 1 to 100, and $\beta$ within 10 to 1000. The dimensions $n$ of the Koopman operator vary within the range of 16 to 2048.
> 3. It can be seen that:
> - Koodos exhibits stable behavior and robust convergence across wide varied hyperparameters, evidencing its ease of training, robust convergence, and insensitivity to hyperparameter.
> - Setting the loss weights in accordance with the magnitude of each loss term is sufficient for achieving good performance.
> - A few hundred Koopman operator dimensions are sufficient good; too many can lead to overfitting.
>
> >Q1. The article repeatedly emphasizes the importance of "C" in CTDG and the arbitrary selection of time t. However, does the proposed method specifically address this aspect?
>
> 1. As we discussed in W1., CTDG extends TDG, relaxing the requirement for fixed-interval discrete-time domain collection and test not limited to immediate one-step future domains. Defining tasks in continuous time is challenging because it breaks the discrete-time models (e.g., state-space models, LSTM) that traditional TDG relies on.
> 2. Therefore, we theoretical proof that the parameter of time-varing model for continuous temporal domains is also continuous, leading to a dynamic predictive model described by differential equations.
> 3. Solving differential equations analytically is almost impossible. The Koopman operator simplifies learning these complex differential equations, enabling effective numerical solutions.
> 4. By the learned differential equations, the parameters of the predictive model can be calculated at any specific time using ODEssolvers.
>
> >Q2. In the methodology section, the article mentions that domain conditional probability distributions will not experience abrupt changes. However, this situation holds true only when the sampling time intervals are equal. The method addresses arbitrary sampling; does this imply it also deals with abrupt changes?
>
> The assumption that domain conditional probability is continues changes is based on the nature of the underlying processes being continuous. The observation intervals, whether equal or arbitrary, do not influence the continuity of the underlying processes themselves. For example, the temperature change throughout a day is a continuous process, regardless of whether measurements are taken every hour or at irregular intervals. The continuity of the temperature change remains inherent to the process itself.

---

> > ### Comment · Reviewer_xaRP · 2024-08-10
> >
> > Thanks for the comprehensive response and my main concerns are well resolved. Thus, I decided to increase my score.

---

> > > ### Author Response · Authors · 2024-08-10
> > > **Response to Reviewer xaRP**
> > >
> > > Thank you for your thoughtful consideration and for recognizing our efforts. Your feedback has been invaluable in refining our work.

---

### Official Review · Reviewer_2Dt9 · 2024-07-13

**Soundness:** 3
**Presentation:** 3
**Contribution:** 3
**Rating:** 6
**Confidence:** 3

**Summary:**

This paper introduces a new task: Continuous Temporal Domain Generalization (CTDG) to address the limitations of traditional TDG in handling continuously evolving and irregularly observed temporal data. By proposing the Koopman operator-driven framework (Koodos), this work leverages Koopman theory and optimization strategies to learn and control the continuous dynamics of data and models. Experiments demonstrate the effectiveness and efficiency of Koodos in managing complex high-dimensional nonlinear dynamics and generalizing across continuous temporal domains.

**Strengths:**

1. This paper introduces a valuable new task, Continuous Temporal Domain Generalization (CTDG), which moves beyond the discrete nature of traditional TDG tasks, enhancing their relevance to real-world applications.

2. The paper proposes the Koopman operator-driven framework (Koodos), effectively leveraging Koopman theory and optimization strategies to understand and control the continuous dynamics of both data and models.

3. It expands the evaluation setting of traditional TDG tasks by introducing datasets with truly continuous data and temporal distribution shifts. This setting encompasses the previous discrete TDG evaluation as a special case, providing a more comprehensive evaluation framework for all TDG methodologies.

4. The authors further evaluate their method and other TDG baselines under this new CTDG evaluation setting, achieving significantly improved performance.

5. The paper is well-written and well-organized.

**Weaknesses:**

1. Evaluation is limited to small datasets and models, lacking assessment on higher-dimensional datasets, which limits the practical applicability of proposed benchmarks and obscures performance in scenarios with large models and high-dimensional data.

2. The continuous nature of CTDG proposed here appears similar to task settings in CIDA and AdaGraph, which focus on domain adaptation. This reduces the novelty of the task setting in this paper, necessitating further discussion and experimental comparisons for clarification.

3. TKNets, another Koopman theory-based TDG method, is relevant but lacks clear differentiation through discussion and experimental comparisons.

4. Including information on the training and inference costs of each method would enhance the completeness of the study.

**Questions:**

1. Does CTDG still adhere to the assumption of smooth distribution shifts?

2. Is the quantitative evaluation in the paper still performed on the last domain?

3. Regarding the Yearbook dataset under CTDG evaluation settings, is it essentially still discrete but can be approximated as continuous due to its large number of domains? If so, which other datasets in CTDG use dense discrete domains to approximate true continuous data?

**Limitations:**

See my weakness part.

---

> ### Author Rebuttal · Authors · 2024-08-07
>
> Thanks for reviewing and acknowledging our work! We see your main concerns are about related works and evaluations. Please read our answers along with the Rebuttal PDF.
>
> > W1. Limited evaluation on small datasets/models; lacks high-dimensional datasets.
> 1. We provide a summary of the datasets and predictive models used in our work in Tab. 5. From it, it is evident that we have utilized high-dimensional datasets and modest-sized datasets and models in our evaluation: the Yearbook dataset comprises around 20k images with 32*32=1024 pixels, and the model used for this dataset has approximately 163k parameters. Similarly, the Cyclone dataset, with image dimensions of 64x64x2=8192, is tested with a model having 135k parameters.
> 2. As a comparison, the existing discrete TDG datasets benchmark only uses data below 100 dimensions, and the minimum size of the model parameter is 12k.
> 3. Combined with the intrinsic challenge in TDG field, further increasing model size to a very large scale (e.g., LLMs) is a promising open area that is considered as a great future direction.
>
> > W2. CTDG's continuous nature resembles CIDA and AdaGraph settings, needing further discussion and comparisons.
> 1. Our work differs from their setup in the following points:
>     - **Different Fields**: CIDA and AdaGraph focus on domain adaptation with access to target domain data during training. CTDG extends TDG and focuses on domain generalization without accessing target domain.
>     - **Different Semantics**: the term 'continuous' is used differently in each work. CIDA: Domain index as a continuous variable instead of a categorical variable to aid the discriminator with a distance-based loss. AdaGraph: Continuous arrival of test data and online adaptation. CTDG: the temporally sequential domains are collected on **continuous-time** instead of fixed interval discrete-time.
>     - **Different Objectives**: CIDA and AdaGraph follow traditional DA/DG settings without modeling temporal evolution dependence like CTDG. Although modeling such temporal patterns has been theoretically shown to have a lower generalization bound [50].
>     - CTDG's novelty lies in its unique requirenment to capturing temporal dependencies and generalizing in continuous-time temporal domains, a challenge not addressed by CIDA, AdaGraph, or existing TDG methods.
> 2. Modifications to CIDA's then can apply to our dataset. We implemented it.  Tab. 4 shows that CIDA cannot deal well with the CTDG problem.
>
> > W3. TKNets need further discussion and comparisons.
> 1. Our work and TKNets have different starting points, frameworks, and contributions.
>     - **Focus and Motivation**: Our work addresses domain evolution over continuous-time, which existing TDG methods, including TKNets, fail to capture effectively as they rely on discrete-time models (e.g., state-space model, LSTM).
>     - **Central Contribution**: Our primary contributions are theoretical proof that the model for continuous temporal domains is also continuous, based on which we construct a dynamic system described by differential equations. The Koopman operator is used to simplify learning complex differential equations. In contrast, TKNets' core contribution is proposing to use the Koopman operator to model the domain state transition matrix, serving a different role from ours.
> 2. We implemented TKNets as a baseline for the CTDG task. Results can be checked in Tab. 4 and it shows TKNets cannot deal well with the CTDG problem.
>
> > W4. Including the training and inference costs would be better.
>
> We have added related experiments; please check Tab 6. Our model strikes a good balance between training time and effectiveness.
>
> > Q1. Does CTDG still adhere to the assumption of smooth distribution shifts?
> 1. Yes, CTDG extends TDG and retains the smooth assumption.
> 2. Moreover, CTDG mitigates two key assumptions of traditional TDG:
>     - train domains are collected at fixed time intervals without missing, e.g., t=1,t=2,t=3
>     - test domains appear only at a time interval in the immediate future, e.g., t=4
> 3. Unlike these assumptions, CTDG allows training domains to be collected at arbitrary times, e.g., t=1.2, t=2.432, t=5.4693…, and adapts to test domains that appear at any point in the future, e.g., t=6.324, t=8.3, t=9.45, not limited to immediate subsequent steps, nor limited to the future one domain.
>
>
> > Q2. Is the quantitative evaluation in the paper still performed on the last domain?
>
> 1. No. As we answered in Q1, one important task of CTDG is to generalize the model to any future point. Therefore, we perform evaluations in multiple test domains arbitrarily and irregularly distributed after the training period.
> 2. For the number of test domains for each dataset, one can refer to the Tab. 5, while the time distribution of these multiple test domains can be found in Figure 6, marked with gray shading.
>
> > Q3. Is the Yearbook dataset discrete but dense as continuous? How about others?
> 1. No. We do not use dense discrete domains to approximate continuous data. All datasets are designed to be consistent with real-world CTDG case situations: (1) Discrete-Time Domains but with Missing Domains; (2) Event-Driven Domains; (3) Streaming Data with Arbitrary Data Collecting Times.
> 2. All three of these scenarios require CTDG. Therefore, the datasets are designed as follows:
>     - The YearBook dataset represents (1). The raw data is collected by years; we randomly sampled 40 years from 83 years to represent the incomplete temporal domain collection process.
>     - The Cyclone dataset represents (2). When each cyclone occurs, the satellite collects a series of images for its entire life cycle, with the date of its occurrence representing a temporal domain time.
>     - Situation 3 is simulated by setting multiple randomly started data collection windows on the tweet stream and the price stream, to create Influenza dataset and House dataset.

---

> > ### Comment · Reviewer_2Dt9 · 2024-08-07
> > **Response to Authors' Rebuttal**
> >
> > I appreciate the authors' rebuttal, which has addressed some of my major concerns. As a result, I'll slightly increase my rating. However, I'm not entirely confident about this evaluation, particularly regarding the assessment of the methodology. I may further reassess after considering the discussions between the authors and other reviewers. I would be grateful if the authors could address the following concerns:
> >
> > ## 1. Necessity and Urgency of Improvements
> >
> > While I acknowledge that the improvements in task setting and problem modeling for TDG are necessary, I question their urgency. Given that most current TDG datasets are relatively toy-like, with low dimensionality and limited to specific scenarios with relatively simple temporal distribution shifts:
> >
> > - Is the main bottleneck in applying TDG to real-world applications still on the temporal aspect?
> > - Will the proposed modeling and methods hold in more complex scenarios?
> >
> > ## 2. Dataset Realism
> >
> > I appreciate the inclusion of Yearbook and Cyclone datasets, which are indeed more realistic than previous TDG datasets. However, I'd like to point out that these datasets are still quite simple in terms of:
> >
> > - Data dimensionality
> > - Dataset size
> > - Distribution shift complexity
> >
> > ## 3. Validation of Improvements
> >
> > While the design intuitions for the improvements seem reasonable and meaningful, I feel the authors haven't used appropriate datasets to motivate and validate the value of these improvements. Using Yearbook as an example:
> >
> > - High school graduation photos are typically taken at a fixed time each year
> > - Is there a significant difference between continuous-time and fixed-interval discrete-time in this context?
> >
> > ## 4. Theoretical Contributions
> >
> > The authors state, "Our primary contributions are theoretical proof that the model for continuous temporal domains is also continuous." However:
> >
> > - Didn't we already have such insights or proofs from DRAIN or GI?
> > - I acknowledge that formally proving this point is a major contribution if previous work hasn't done so.
> >
> > ## 5. Comprehensiveness of Comparisons
> >
> > Regarding comparisons with other TDG methods:
> >
> > - Are the comparisons comprehensive?
> > - Could we improve the performance of these methods in the CTDG setting by dividing them into denser discrete temporal domains?

---

> > > ### Author Response · Authors · 2024-08-09
> > > **Response to Reviewer 2Dt9 Comments**
> > >
> > > Thank you for your continued engagement and for acknowledging the improvements! We appreciate your thoughtful feedback.
> > > > D1.
> > > > - Is the main bottleneck of TDG application on the temporal aspect?
> > > > - Will the proposed methods hold in more complex scenarios?
> > >
> > > 1. The temporal aspect remains a significant challenge in the field of TDG. Existing TDG models struggle with flexibility in terms of times, e.g., domains in arbitrary times, and domains in further future (than just the next time point), revealing **a critical gap in the ability to abstract accurate temporal dynamics from temporal domain data**. However, our approach alleviates this challenge by leveraging model dynamic systems and Koopman theory, representing a significant advancement in capturing the temporal dynamics of TDG and marking an important research direction worthy of further exploration.
> > > 2. Our methods hold in more complex scenarios. Real-world changes are often driven by continuous processes and the domains of interest can be at arbitrary times, which align with our approach’s foundational assumptions. The continuous nature of these changes means that our methods remain robust even as scenarios increase in complexity.
> > >
> > > > D2. Dataset Realism
> > > 1. Thank you for acknowledging our use of the Yearbook and Cyclone datasets as a step towards more realistic TDG datasets. We appreciate your recognition of our improvements in this area.
> > > 2. We fully acknowledge that larger, higher-dimensional datasets would provide stronger empirical validation. While TDG/CTDG is still an emerging field with developing benchmarks, our endeavors are aimed to push this community forward progressively towards larger scale, more realistic data via this and subsequent works.
> > >
> > > > D3. Yearbook taken at a fixed time each year. Is there difference between continuous-time and discrete-time of such?
> > > 1. The factor in choosing between CTDG and traditional TDG models is not whether data is collected at integer time points, but **whether the time intervals between data collections are fixed or not**. Traditional TDG models require equal intervals because they don't account for time duration amount between consecutive domains. Therefore, when the time intervals are variable, TDG models are blind to such variation. In contrast, continuous-time models consider these durations between consecutive domains and hence can handle domains at arbitrary time points.
> > > 2. The Yearbook dataset used in the study samples 40 years over an 83-year span, introducing variability in time intervals. Traditional TDG models cannot capture such variability, while CTDG models can. It is a significant difference that should not be ignored.
> > >
> > > > D4. Do we have continuous proofs from DRAIN or GI? If no, proving this is a major contribution.
> > >
> > > Thank you for acknowledging our theoretical contribution.
> > > 1. The GI introduced the intuitive idea that encouraging learning functions that can be approximated linearly may improve a model's generalization ability. However, **this was demonstrated through empirical experiments without theoretical proof**.
> > > 2. Moreover, while GI enforces local linearity properties of the model, it does not explore the exact mechanisms of how future model parameters can be determined. In contrast, our Theorem I provides a deep understanding of how CTDG models can be computed and generalized.
> > > 3. DRAIN suggests that future model parameters are conditional on parameter statuses from historical domains, **but it does not assume or require continuity in these parameters**.
> > >
> > > > D5.
> > > > - Are the comparisons comprehensive?
> > > > - Could we improve TDG model in CTDG by dividing them into denser discrete temporal domains?
> > > 1. We have compared widely with well-recognized TDG methods GI, DRAIN, and TKNets using 12 datasets in both continuous and discrete domains under 3 metrics. Our method shows significantly improved performance in all these comprehensive comparisons, clearly demonstrating its effectiveness.
> > > 2. TDG methods underperform in CTDG settings because they can't sense the variability of time duration amount between consecutive domains, which is an important addition of CTDG over TDG.
> > > 3. If we turn CTDG tasks into TDG tasks, we have to interpolate additional domains for the intermediate time points between the time points in CTDG domains. For example, if CTDG domains occur at times (1.1, 2.3, and 5.1), TDG would require interpolating domains at (1.1,1.5,1.9,2.3,2.7,3.1,3.5,3.9,4.3,4.7,5.1) in order to form fixed interval for TDG to apply. Generating the entirety of the data in the domain of each time point is arbitrarily challenging especially considering the size, dimensionality, and complexity of the data is nontrivial and can be larger and larger. And such interpolation will introduce extra uncontrollable error. Moreover, this interpolation also significantly reduces efficiency, potentially making the process impractically slow, as the number of domains could multiply drastically.

---

### Author Rebuttal · Authors · 2024-08-07

Thank you to each reviewer for their valuable time.
Please download the pdf file before reviewing the responses.

Best,
Authors

---

### Decision · Program_Chairs · 2024-09-25

**Decision:**

Accept (poster)

**Comment:**

This paper focuses on continuous domain generalization. All reviewers are in favor of acceptance, although it is important for the authors to incorporate the requested improvements to the paper.